# Lipid droplet-associated hydrolase mobilizes stores of liver X receptor sterol ligands and protects against atherosclerosis

Young-Hwa Goo[1] ✉, Janeesh Plakkal Ayyappan[1], Francis D. Cheeran[1], Sushant Bangru [2,3,4], Pradip K. Saha[5], Paula Baar[6], Sabine Schulz [6], Todd A. Lydic[7], Bernhard Spengler[6,8], Andreas H. Wagner [9], Auinash Kalsotra [2,3,4,10], Vijay K. Yechoor [11] & Antoni Paul[1] ✉

Foam cells in atheroma are engorged with lipid droplets (LDs) that contain esters of regulatory lipids whose metabolism remains poorly understood. LD-associated hydrolase (LDAH) has a lipase structure and high affinity for LDs of foam cells. Using knockout and transgenic mice of both sexes, here we show that LDAH inhibits atherosclerosis development and promotes stable lesion architectures. Broad and targeted lipidomic analyzes of primary macrophages and comparative lipid profiling of atheroma identified a broad impact of LDAH on esterified sterols, including natural liver X receptor (LXR) sterol ligands. Transcriptomic analyzes coupled with rescue experiments show that LDAH modulates the expression of prototypical LXR targets and leads macrophages to a less inflammatory phenotype with a profibrotic gene signature. These studies underscore the role of LDs as reservoirs and metabolic hubs of bioactive lipids, and suggest that LDAH favorably modulates macrophage activation and protects against atherosclerosis via lipolytic mobilization of regulatory sterols.

The etiology of atherosclerosis is intrinsically linked to lipid deposition and chronic inflammation at the arterial wall[1]. A large proportion of lipids in atheroma are stored in the cytoplasmic lipid droplets (LDs) of macrophages that are commonly known as foam cells. Lipid storage often requires esterification, and mobilization of these stores involves ester hydrolysis. Since lipoproteins that infiltrate the arterial wall and are engulfed by foam cells carry abundant cholesterol, foam cells are characteristically rich in cholesterol and cholesterol esters (CEs), and the mechanisms by which cholesterol is trafficked through LDs have been extensively studied. However, the lipidome of foam cells and atherosclerotic lesions is highly heterogeneous, and it is increasingly recognized that LDs also function as central reservoirs and metabolic sites for bioactive lipids[2–6]. The molecular mechanisms that govern the metabolism of some of these lipids, including sterol metabolites that are potent regulators of foam cell biology and atherogenesis, remain poorly understood.

The metabolism of LDs is regulated by associated proteins of diverse functions[7–9]. Lipid droplet-associated hydrolase (LDAH) is a newly identified LD protein that is highly expressed by macrophages within atherosclerotic lesions of mice and humans[10]. It contains a

[1]Department of Molecular and Cellular Physiology, Albany Medical College, Albany, NY, USA. [2]Department of Biochemistry, University of Illinois, Urbana-Champaign, IL, USA. [3]Cancer Center@Illinois, University of Illinois, Urbana-Champaign, IL, USA. [4]Carl R. Woese Institute for Genomic Biology, University of Illinois, Urbana-Champaign, IL, USA. [5]Department of Molecular and Cellular Biology, Baylor College of Medicine, Houston, TX, USA. [6]Institute of Inorganic and Analytical Chemistry, Justus Liebig University Giessen, Giessen, Germany. [7]Department of Physiology, Michigan State University, East Lansing, MI, USA. [8]TransMIT GmbH, Center for Mass Spectrometric Developments, Giessen, Germany. [9]Department of Cardiovascular Physiology, Heidelberg University, Heidelberg, Germany. [10]Division of Nutritional Sciences, University of Illinois, Urbana-Champaign, IL, USA. [11]Division of Endocrinology, Diabetes, and Metabolism, Department of Medicine, University of Pittsburgh, Pittsburgh, PA, USA. ✉e-mail: gooy@amc.edu; paula@amc.edu

conserved canonical lipase/esterase motif and has a high affinity for LDs, although its roles in lipid metabolism and atherosclerosis development have not yet been resolved[10–14]. In this report we leverage LDAH transgenic and knockout mouse models to investigate LDAH's lipid substrates and roles in foam cell biology and atherogenesis. We show that LDAH protects against atherosclerosis and promotes more benign plaque phenotypes, characterized by increased deposition of fibrillar collagen and reduced areas of necrosis. Unbiased global and targeted lipidomic analyses of primary macrophages and atherosclerotic lesions identify sterol esters as candidate LDAH substrates, including bioactive sterols. Transcriptomic analyses show that LDAH induces expression of prototypical liver X receptor (LXR) target genes and promotes an alternatively activated macrophage phenotype with a profibrotic molecular signature. Collectively, these studies identify a novel player in the lipolytic release of regulatory sterol depots and link this metabolic step to favorable modulation of foam cell phenotype and protection against atherosclerosis development.

## Results

### Myeloid LDAH protects against atherosclerosis and promotes stable lesion architectures

To investigate how LDAH gain-of-function affects atherosclerosis development, we generated myeloid-specific LDAH transgenic (LDAH-Tg) mice under the control of the promoter and enhancer of the mouse *Csf1r* locus, which direct position and copy number-independent expression in macrophages and granulocytes (Fig. 1A)[15,16]. Increased LDAH expression in macrophages from hemizygous transgenics (*Ldah*<sup>Tg/0</sup>) was confirmed at the RNA and protein levels (Fig. 1B, C). *Ldah*<sup>Tg/0</sup> mice were crossed with *Apoe*<sup>−/−</sup> mice to generate *Ldah*<sup>Tg/0</sup>*Apoe*<sup>−/−</sup> and *Ldah*<sup>0/0</sup>*Apoe*<sup>−/−</sup> littermates. After 20 weeks under standard chow, body weight, plasma triglyceride, total and HDL cholesterol, and the overall cholesterol distribution among lipoprotein fractions were similar between genotypes in both sexes (Table S1 and Supplementary Fig. S1). However, atherosclerosis development was significantly reduced in *Ldah*<sup>Tg/0</sup>*Apoe*<sup>−/−</sup> mice, both in male and female mice, indicating that LDAH is an atheroprotective player (Fig. 1D, E).

Histological analyses also revealed important differences in lesion composition. Lesions of female mice of both genotypes consisted mainly of advanced fibroatheroma. Compared to their WT littermates, the lesions of *Ldah*<sup>Tg/0</sup>*Apoe*<sup>−/−</sup> females displayed a less vulnerable phenotype, with a marked reduction (> 50%) in apoptotic cells and necrotic areas, slightly reduced Mac3 positive areas, and increased collagen deposition (Fig. 2A). As compared to the females, the males of both genotypes developed less advanced lesions (Fig. 2B). In both genotypes the lesions consisted primarily of Mac3 positive fatty streaks, and there were no differences between genotypes in the percent of lesion area that stained positive for Mac3 (Fig. 2B). Necrotic cores were still scarce in males of both genotypes, and there were no differences between genotypes (Fig. 2B). However, the number of apoptotic cells was already significantly reduced in *Ldah*<sup>Tg/0</sup>*Apoe*<sup>−/−</sup> males (Fig. 2B). Remarkably, while at this stage of development lesions of *Ldah*<sup>0/0</sup>*Apoe*<sup>−/−</sup> male mice still contained little collagen, the lesions of *Ldah*<sup>Tg/0</sup>*Apoe*<sup>−/−</sup> males were markedly fibrotic, and quantitative analysis of sections stained with Masson's trichrome showed a ~ 3-fold increase in collagen deposition (Fig. 2B). Type I fibrillar collagen is normally the main extracellular matrix (ECM) component in atheroma, where it plays a pro-stabilizing role by increasing the lesion's tensile strength[17]. Immunostaining with an anti-type I collagen antibody confirmed that lesions of transgenic mice are richer in type I collagen fibers (Supplementary Fig. S2).

For loss-of-function studies, we generated LDAH knockout (KO, *Ldah*<sup>−/−</sup>) mice by homologous recombination (Supplementary Fig. S3) and bred them into *Apoe*<sup>−/−</sup> background. Body weight and plasma lipids were similar between *Ldah*<sup>+/+</sup>*Apoe*<sup>−/−</sup> and *Ldah*<sup>−/−</sup>*Apoe*<sup>−/−</sup> littermates in both sexes (Table S2 and Fig. S4). In mice fed regular chow, global LDAH deficiency or LDAH deficiency in bone marrow-derived cells did not affect lesion size (Supplementary Fig. S5). However, in line with the findings in transgenic mice (Fig. 2), lesions of LDAH-KO female mice contained less collagen and larger areas of necrosis than lesions of WT littermates (Supplementary Fig. S5). Thus, we investigated how loss of LDAH affects atherosclerosis in mice challenged with a western diet (WD). There were no differences in body weight and plasma lipids

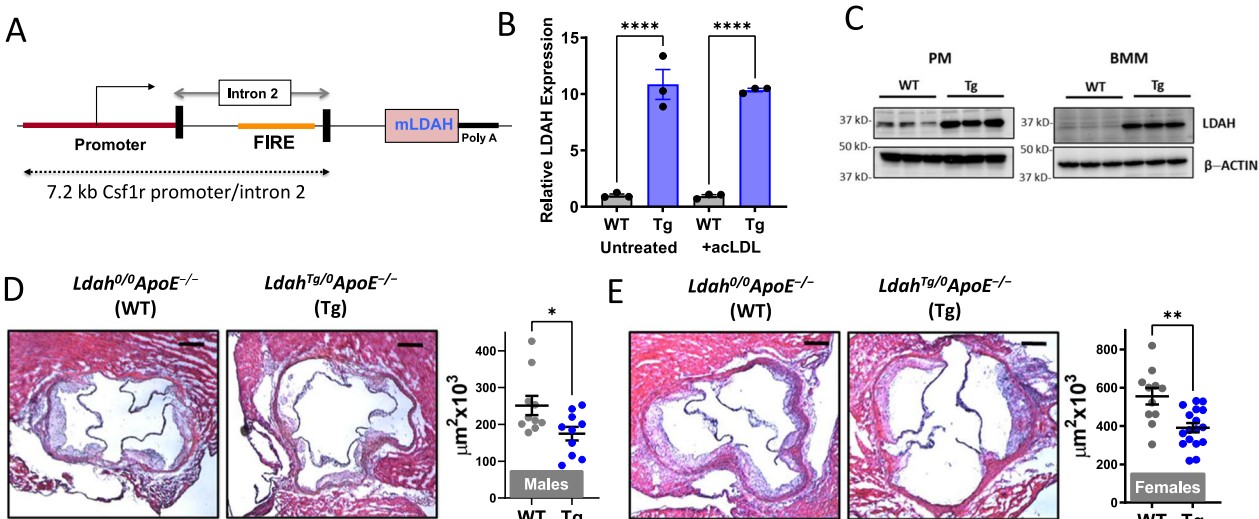

**Fig. 1 | LDAH overexpression protects against atherosclerosis. A** LDAH transgene design with the colony stimulating factor 1 receptor (Csf1r) promoter and intron 2 that contains the enhancer element FIRE. **B** LDAH RNA expression in LDAH-Tg (Tg, blue bars) and LDAH-WT (WT, gray bars) peritoneal macrophages (PM) that remained untreated or were treated with acetylated LDL (acLDL, 50 μg/ml). Data are presented as mean ± SEM of 3 independent samples. ****$P < 0.0001$, two-way ANOVA followed by Tukey's test. **C** Western blot showing increased LDAH protein in transgenic peritoneal macrophages (PM) and bone marrow-derived macrophages (BMM). The panel shows $n = 3$ biological replicates representative of at least 3 independent experiments. **D, E** Representative H&E images of aortic root sections and quantification of atherosclerotic lesion size in 20-week-old male (**D**) and female (**E**) *Ldah*<sup>0/0</sup>*ApoE*<sup>−/−</sup> (WT, $n = 10$ male and 11 female, gray dots) and *Ldah*<sup>Tg/0</sup>*ApoE*<sup>−/−</sup> (Tg, $n = 10$ male and 17 female, blue dots) littermates fed standard chow. Data are presented as mean ± SEM. All bars= 200 μm. *$P = 0.0433$ by two-tailed Mann-Whitney U test (panel D); **$P = 0.0013$ by two-tailed unpaired *t*-test (**E**). Source data are provided as a Source Data file.

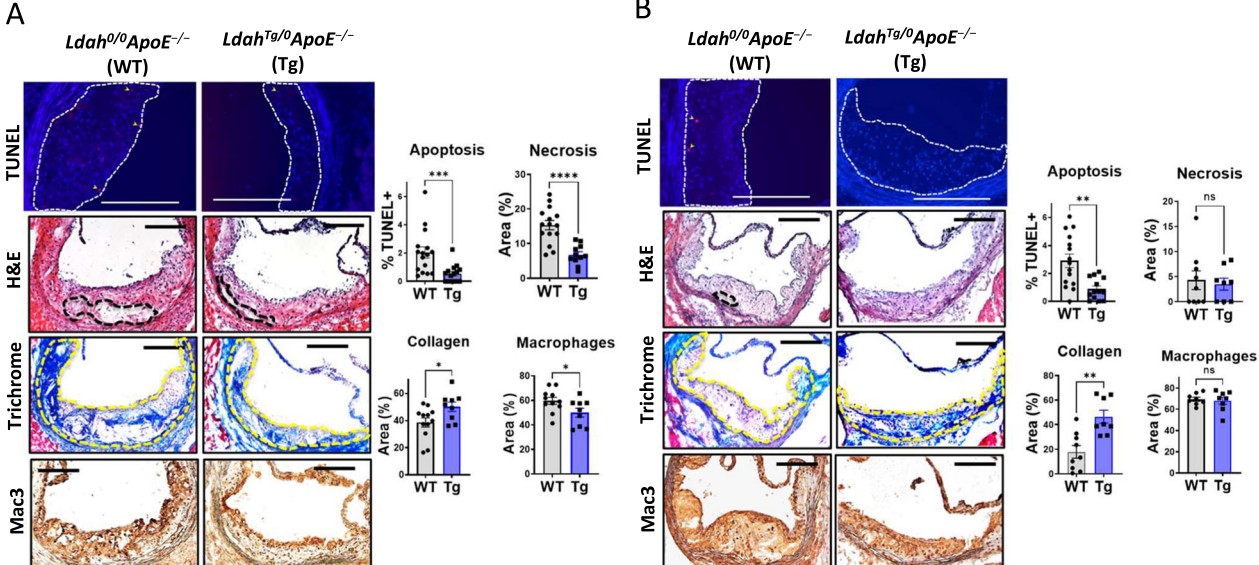

**Fig. 2 | LDAH overexpression promotes less necrotic and highly fibrotic plaque phenotypes.** Representative images and quantification of TUNEL+ cells, necrotic areas (outlined by dashed lines) in H&E staining, Mac3 staining (brown) and collagen (blue in trichrome staining) in lesions of *Ldah⁰/⁰ApoE⁻/⁻* (WT, gray bars) and *Ldah^Tg/0^ApoE⁻/⁻* (Tg, blue bars) female (**A**) and male (**B**) littermates. Apoptotic cells identified by TUNEL are indicated by arrows. Mac3+ areas within lesions are stained brown. Lesion areas in trichrome staining are outlined in yellow. All bars = 200 μm. Data are presented as mean ± SEM independent samples. Apoptosis (female *n* = 16 WT and 15 Tg, ***P* = 0.0008; male *n* = 15 WT and 13 Tg, ***P* = 0.0011); necrosis (female *n* = 15 WT and 11 Tg, *****P* < 0.0001*; male *n* = 9 WT and 8 Tg, n.s. not significant); collagen (female *n* = 12 WT and 9 Tg, **P* = 0.0302; male *n* = 9 WT and 8 Tg, ***P* = 0.0037); macrophages (female *n* = 11 WT and 9 Tg, **p* = 0.0468; male *n* = 8 WT and 8 Tg, n.s. not significant). Comparisons were performed using two-tailed unpaired *t*-test (Mac3 in male and female, necrosis collagen and TUNEL in female), two-tailed Welch's *t*-test (TUNEL in male) or two-tailed Mann-Whitney U test (necrosis and collagen in male). Source data are provided as a Source Data file.

between genotypes (Supplementary Table S2 and Supplementary Fig. S4). However, after 12 weeks of WD feeding *Ldah⁻/⁻Apoe⁻/⁻* mice of both sexes developed significantly larger lesions than their *Ldah⁺/⁺Apoe⁻/⁻* littermates (Fig. 3A, B). Bone marrow transfer (BMT) experiments also showed enhanced lesion development in *Ldah⁻/⁻Apoe⁻/⁻* bone marrow-engrafted mice (Fig. 3C). Phenotypically, the lesions of LDAH-KO mice contained more apoptotic cells, larger necrotic cores, increased Mac3 positive areas, and reduced collagen (Fig. 3D–G). The effects of LDAH deficiency on lesion composition were consistent in both sexes, and LDAH deficiency in bone marrow-derived cells also phenocopied the effects of global deficiency on lesion composition (Fig. 3D–G).

Altogether, atherosclerosis studies performed under both LDAH deficiency and overexpression identify LDAH as an atheroprotective player that promotes favorable lesion remodeling by reducing necrosis and increasing fibrosis, and suggest that the mechanisms of atheroprotection involve modulation of myeloid cells.

## LDAH mobilizes stores of regulatory sterols

Given LDAH's lipase structure and association with LDs, to investigate the molecular mechanisms behind LDAH's atheroprotective functions we first profiled the lipidome of oxidized LDL (oxLDL)-treated peritoneal macrophages (PM) using shotgun lipidomics. There were no significant differences in total lipid content between LDAH-KO and LDAH-Tg PM and their respective WT controls (Fig. 4A, B). However, LDAH selectively impacted the levels of sterol lipids and cardiolipins (CL), which were significantly reduced in LDAH-Tg PM and significantly increased in KDAH-KO PM (Fig. 4A, B and Supplementary Tables S3, S4). Lipid biosynthesis primarily takes place in the endoplasmic reticulum (ER), and LDs that emerge from the ER serve as reservoirs for excess lipids that are generated in or trafficked through the ER[8]. However, CL is the signature lipid of mitochondrial membranes and is synthesized in and almost exclusively localized in the inner mitochondrial membrane, which makes it an unlikely direct target of a LD-associated enzyme[18]. In addition, although LDAH reduced the total levels of CL, its effects

were not uniform among all CL species (Supplementary Fig. S6). Conversely, sterol esterification takes place in the ER, and foam cell LDs serve as reservoirs of abundant esterified sterols[2,3]. Regardless of the type of fatty acid esterified to the sterol, LDAH's impact was very consistent across all individual CE species identified in the lipidomic analyzes (Fig. 4C). Most CE species were elevated in LDAH-KO macrophages, while in LDAH-Tg macrophages the levels of all CEs detected were lower than in their WT counterparts, and all differences were statistically significant (Fig. 4C).

The ER-resident enzyme acyl-coenzyme A:cholesterol acyltransferase-1 (ACAT1) that esterifies cholesterol also esterifies other sterol substrates, including side chain oxysterols and cholesterol synthesis intermediates that are natural LXR ligands[19–21]. In foam cells and atheroma some of these sterols are abundantly found as esters, suggesting that they could also be substrates of LDAH[3,4,6,22,23]. Thus, we further profiled sterol lipids using high-resolution LC-MS. The sterol panels also revealed a robust impact of LDAH across different sterol species. While the differences in cholesterol were modest, the total levels of other metabolites were significantly higher in LDAH-KO PM and significantly lower in LDAH-Tg PM than in their respective WT controls (Fig. 5A, C). We detected multiple sterols that are present in atheroma, including characteristic oxidized products at the seventh carbon of the sterol ring system, such as 7-ketocholesterol (7-KC) and 7α-hydroxycholesterol (7α-HC), side-chain oxysterols such as 25-hydroxycholesterol (25-HC) and 27-hydroxycholesterol (27-HC), and desmosterol. Most of these metabolites were elevated in LDAH-KO PM, and in LDAH-Tg macrophages all metabolites were significantly lower than in WT controls (Fig. 5B, D).

Esterification and compartmentalization in hydrophobic compartments could significantly limit the accessibility and function of regulatory sterols and, as with cholesterol, the rate of ester hydrolysis was shown to be a main determinant of the homeostasis of 25-HC[23,24]. While it is established that many of the sterols present in atheroma are esterified by ACAT1, the mechanisms and consequences of ester hydrolysis remain poorly understood[19,20]. We designed trafficking

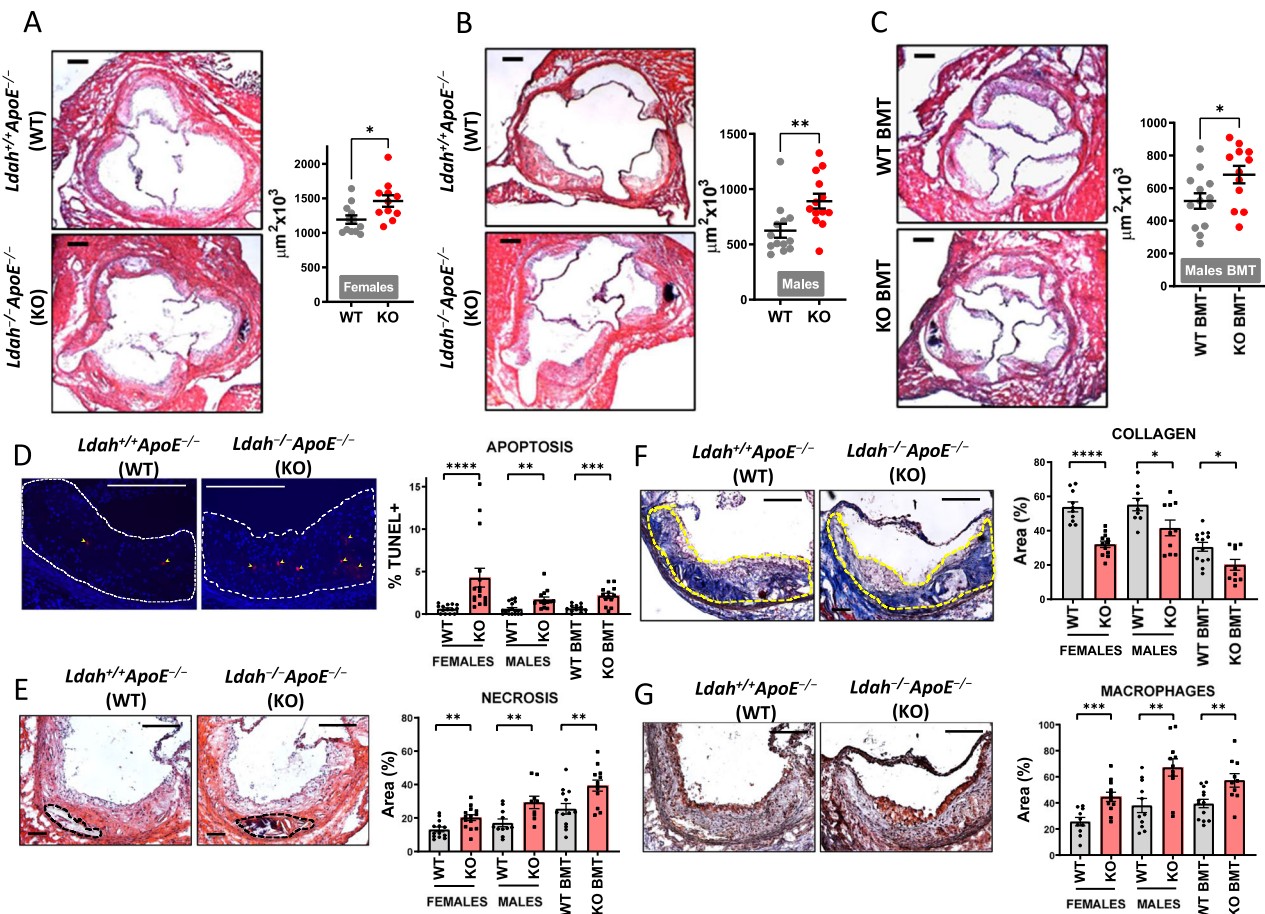

**Fig. 3 | LDAH deficiency increases atherosclerosis development and decreases plaque stability. A, B** Global LDAH deficiency. At 8 weeks of age mice were switched from regular chow to WD for 12 weeks, until age 20 weeks. Images and charts show representative H&E images of aortic root sections and quantification of atherosclerotic lesion size in *Ldah*⁺ᐟ⁺*Apoe*⁻ᐟ⁻ (WT, gray dots) *vs. Ldah*⁻ᐟ⁻*Apoe*⁻ᐟ⁻ (KO, red dots) female (n = 12 WT and 11 KO, *P = 0.0154) (**A**) and male (n = 13 WT and 12 KO, **P = 0.0051) (**B**) littermates. **C** Bone-marrow specific LDAH deficiency. Eight-week-old *Apoe*⁻ᐟ⁻ mice were transplanted with bone-marrow cells isolated from *Ldah*⁻ᐟ⁻*Apoe*⁻ᐟ⁻ mice (KO BMT, gray dots, n = 12) or *Ldah*⁺ᐟ⁺*Apoe*⁻ᐟ⁻ mice (WT BMT, red dots, n = 13) *P = 0.0309. Mice were fed regular chow for an additional 4 weeks after transplant and at 12 weeks of age mice were switched to a WD for 12 additional weeks. **D** Representative images and quantification of TUNEL+ cells (pointed by arrows) (n = 14 WT and 16 KO females, ****P < 0.0001; 17 WT and 14 KO males, **P = 0.0018; 12 WT and 13 KO BMT recipients, ***P = 0.0004). **E** Representative H&E images with necrotic areas outlined by black dashed line, and quantification of necrosis (n = 13 WT and 14 KO females, **P = 0.005; 11 WT and 9 KO males,

**P = 0.0087; 13 WT and 12 KO BMT recipients, **P = 0.0064). **F** Representative Masson's Trichrome images with lesion areas outlined in yellow, and quantification of collagen (blue) (n = 10 WT and 12 KO females, ****P < 0.0001; 9 WT and 10 KO males, *P = 0.0346; 13 WT and 10 KO BMT recipients, *P = 0.0121). **G** Representative immunostaining with anti-Mac3 (brown), and quantification of Mac3-positive areas (n = 10 WT and 12 KO females, ***P = 0.0009; 11 WT and 11 KO males, **P = 0.0029; 13 WT and 10 KO BMT recipients, **P = 0.0062). All representative images in **D–G** correspond to lesions of WT females *vs.* females with LDAH deficiency. All gray bars in Figures D-F represent WT mice, and red bars represent KO mice. All scale bars: 200 μm. Data are presented as mean ± SEM of independent samples. Comparisons were performed by two-tailed Mann-Whitney U test for male lesions (panel B), apoptosis in male and female mice, and collagen in BMT mice. Two-tailed Welch's *t*-test was used for apoptosis in BMT mice. All other comparisons were performed using two-tailed unpaired *t*-test. Source data are provided as a Source Data file.

experiments to assess the esterification and hydrolysis of two of the sterols affected by LDAH, cholesterol and 25-HC, following a protocol similar to that described by Venkateswaran et al.[25]. LDAH-Tg and WT peritoneal macrophages were incubated with a mixture of acetylated LDL (acLDL) and 25-HC, and subsequently pulsed with ³H-oleate to specifically label esters formed de novo intracellularly. To assess the rates of ester hydrolysis, cells were chased in the presence of apolipoprotein A-I (apoA-I) under ACAT1 inhibition (Fig. 5E). Following the treatment with ³H-oleate there were no differences in ³H-labeled CE or 25-HC ester (25-HCE) between genotypes, indicating that LDAH does not affect ester formation (Fig. 5F). However, the turnover of both esters was significantly faster under LDAH overexpression, and 6 h after incubation with apoA-I, LDAH-Tg PM contained ~30% less CE than WT PM and displayed an even more marked reduction (~70%) in 25-HCE (Fig. 5G). Taken together, lipidomics and trafficking experiments suggest that LDAH affects the homeostasis of multiple

sterols by facilitating ester mobilization, including important bioactive metabolites.

## LDAH modulates the foam cell's transcriptome through an LXR-dependent mechanism

Lipidomic studies suggest that LDAH mobilizes stores of sterols that are endogenous LXR ligands. Thus, we asked how LDAH gain- and loss-of-function affect the expression of two prototypical LXR target genes: the ATP-binding cassettes *Abca1* and *Abcg1.* As seen in Fig. 6A, in oxLDL- treated PM the expression of both genes was significantly induced and reduced under LDAH overexpression and deficiency, respectively. To further investigate the effects of LDAH on the foam cell's transcriptome, we performed RNA-sequencing (RNA-seq) analyses. Comparisons between LDAH-Tg and WT PM treated with oxLDL identified 872 differentially expressed genes (DEGs) at an FDR-adjusted p-value (q-value) of <0.05 (Supplementary Table S5 and

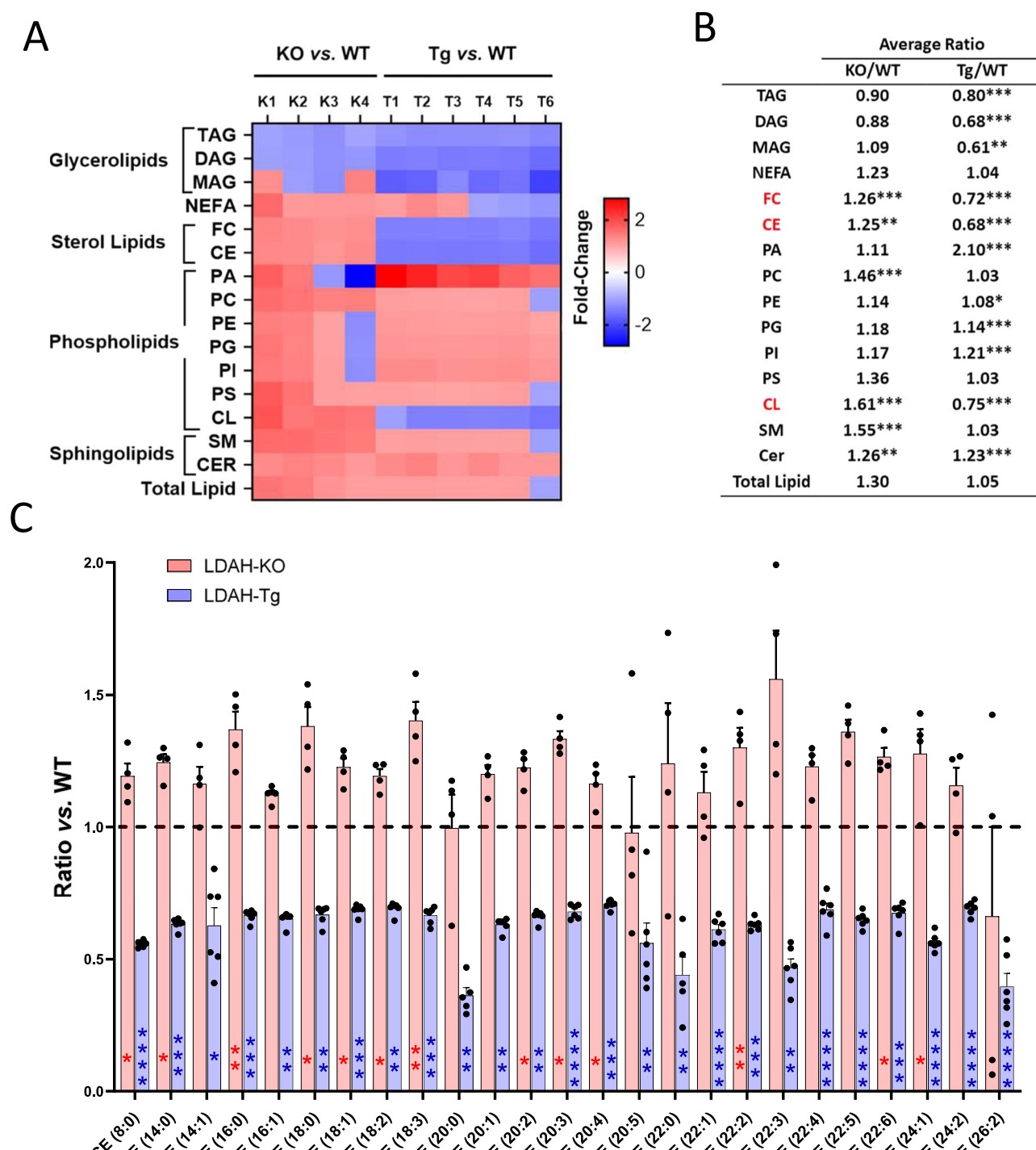

**Fig. 4 | Shotgun lipidomics profiling of LDAH's impact on the foam cell's lipidome. A**, **B** Summary of the effects of LDAH deficiency and overexpression on the lipidome of PM treated with oxLDL (50 µg/ml) for 48 h. **A** Heat map. Each column represents the ratios of independent LDAH-KO (K) and LDAH-Tg (T) samples over the average of their respective wild-type (WT) controls. **B** Average lipid ratios in LDAH-KO ($n = 4$) *vs.* WT ($n = 5$) controls (KO/WT) and LDAH-Tg ($n = 6$) *vs.* WT controls ($n = 6$) (Tg/WT). Lipids significantly and inversely regulated under LDAH deficiency and overexpression are shown in red. **C** Effects of LDAH on individual cholesterol ester species. Red bars represent the mean ratio ± SEM of LDAH-KO ($n = 4$) over WT control ($n = 5$) independent samples. Blue bars represent the ratio of LDAH-Tg ($n = 6$) over WT ($n = 6$) control. The first number in the parenthesis represents the number of carbons of the fatty acid esterified to the sterol, and the number after the colon represents the number of double bonds. *$P < 0.05$, **$P < 0.01$, ***$P < 0.001$, ****$P < 0.0001$ by two-tailed unpaired *t*-test (normally distributed with equal variances), two-tailed Welch's *t*-test (normally distributed with unequal variances), or two-tailed Mann-Whitney U (not normally distributed). CE cholesterol ester, CER ceramide, CL cardiolipin, DAG diacylglycerol, FC free cholesterol, MAG monoacylglycerol, PA phosphatidic acid, PC phosphatidylcholine, PE phosphatidylethanolamine, PG phosphatidylglycerol, PI phosphatidylinositol, PS phosphatidylserine, SM sphingomyelin, TAG triacylglycerol. Source data are provided as a Source Data file.

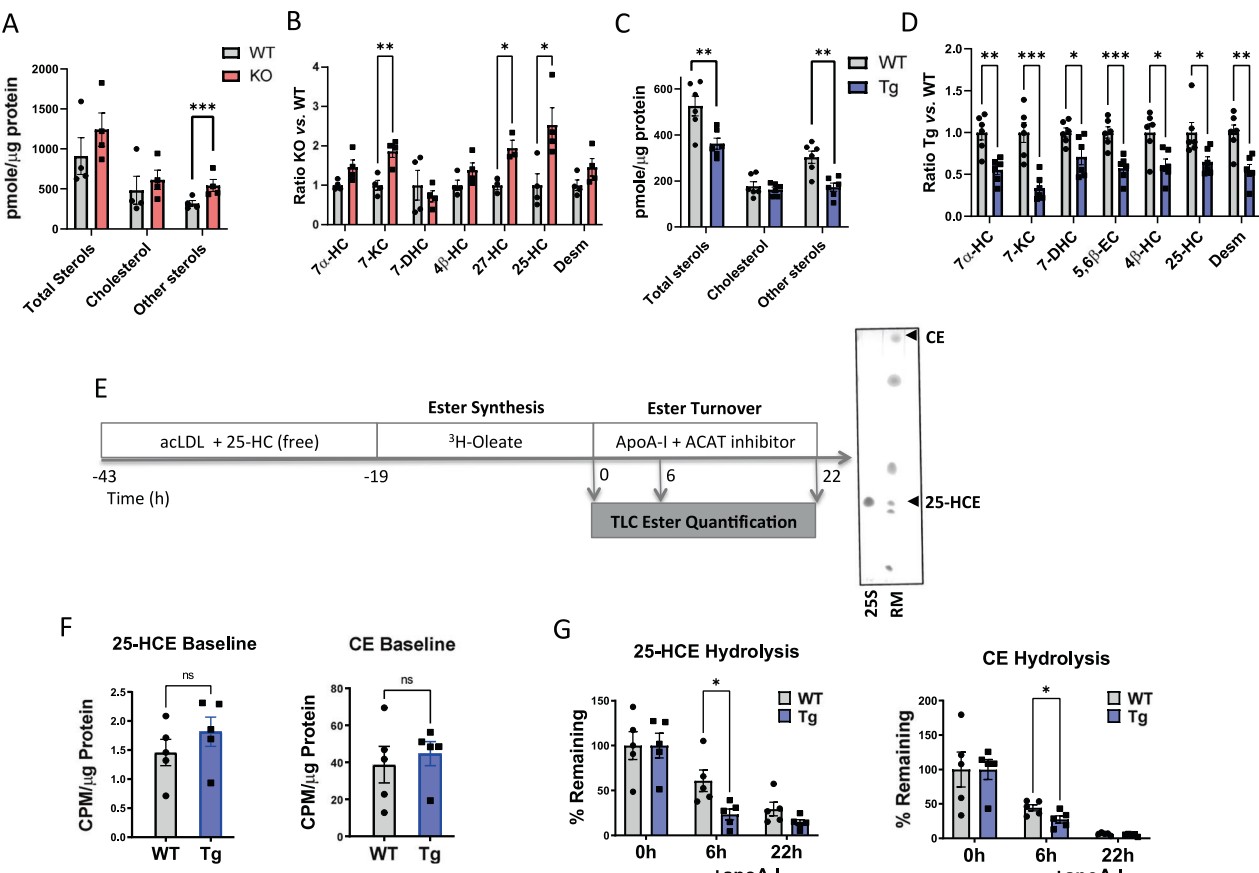

**Fig. 5 | LDAH mobilizes oxysterol stores. A–D** PM isolated from LDAH-Tg (Tg) and LDAH-KO (KO) mice and their respective WT controls were treated with oxLDL (50 µg/ml) for 48 h and lipid extracts were analyzed for sterols using LC-MS/MS (n = 4-6 independent samples). **A** Levels of total sterols, cholesterol, and oxysterols and (**B**) relative levels of the main oxysterol species identified in LDAH-KO PM (red bars) with respect to WT control (gray bars) PM (n = 4). **C** Levels of total sterols, cholesterol and oxysterols and (**D**) relative levels of the main oxysterol species identified in LDAH-Tg (blue bars) with respect to WT (gray bars) control PM (n = 6). 7α-HC = 7α-Hydroxycholesterol; 7-KC = 7-ketocholesterol; 5,6ß-EC = 5,6ß-Epox-ycholesterol; 4ß-HC = 4ß-hydroxycholesterol; 25-HC = 25-hydroxycholesterol; 27-HC = 27-hydroxycholesterol; Desm= desmosterol. **E** Design of trafficking

experiments to determine the rates of esterification and ester hydrolysis of cho-lesterol and 25-HC. Purified 25-HC oleate (25 S) and a TLC standard reference mixture (RM) were used to identify the 25-HCE and CE bands in TLC analysis. **F** 25-HC ester (25-HCE) and cholesterol ester (CE) levels after a pulse with [3]H-oleate (baseline)(n = 5). **G** 25-HC and CE levels following chase with apoA-I (hydrolysis) (n = 5) Gray bars in F and G represent WT PM and blue bars correspond to LDAH-Tg PM. All bars represent mean ± SEM of independent samples. 25-HC in (**D**) total sterols in panel A and CE in panel F were analyzed using two-tailed Mann-Whitney U test. All other comparisons were performed using two-tailed unpaired *t*-test. *P < 0.05, **P < 0.01, ***P < 0.001. Source data are provided as a Source Data file.

Supplementary Fig. S7A). In turn, comparisons between LDAH-KO and WT controls identified 87 DEGs at q < 0.05 (Table S6 and Fig. S7B). Of nine common targets between the two RNA-seq analyses, four were reciprocally regulated by LDAH deficiency and overexpression, including the *Col1a1* gene, which was upregulated in LDAH-Tg PM and downregulated in LDAH-KO PM (Fig. 6B). We next looked for genes that changed at q < 0.05 in either Tg or KO macrophages and were reciprocally regulated in the opposite genotype. Among 246 genes upregulated by LDAH, there were multiple players in collagen home-ostasis (Fig. 6C and Supplementary Table S7), and gene ontology enrichment analysis of this gene list returned several terms related to extracellular matrix (ECM) synthesis and organization (Supplementary Fig. S8). Using qPCR we confirmed that the *Col1a1* and *Col1a2* genes that encode the two main chains of type I collagen fibers were upre-gulated and downregulated in LDAH-Tg and LDAH-KO macrophages, respectively (Fig. 6D)[17]. By ELISA, we also found increased levels of pro-collagen I-alpha in LDAH-Tg BMM treated with oxLDL (Fig. 6F). These profibrotic changes align with the increased collagen deposition we have consistently observed in atherosclerosis studies.

LDAH transgenic overexpression significantly (q < 0.05) affec-ted the expression of 98 genes identified in previous studies that

performed genome-wide ChIP-seq and transcript profiling in mac-rophages to identify genes that are activated and repressed by LXR agonizts (Table S8)[26,27]. However, although qPCR analyzes of PM had found *Abca1* and *Abcg1* upregulated by LDAH, these two genes were not identified in the RNA-seq analyzes. By Western blotting, we confirmed that ABCA1 protein is higher in BMM treated with oxLDL (Supplementary Fig. S9). To further investigate whether LDAH induces *Abca1* and *Abcg1* in an LXR-dependent manner, we knocked down LXRα and LXRβ using pooled siRNA. Some studies using free oxysterols or synthetic LXR agonizts have shown redundancy between LXRα and LXRβ in *Abca1* expression, while others, including independent atherosclerosis studies where LXR activa-tion is primarily driven by endogenous ligands, did not see LXRβ compensation under LXRα deficiency, suggesting that there might be differences in responses depending on experimental variables such as the dose, type, or source of the agonizts[25,28–31]. In oxLDL-treated BMM, LXRα knockdown alone reduced the expression of *Abca1*, while LXRβ knockdown had no effects (Supplementary Fig. S10). As in PM, the induction of *Abca1*, *Abcg1* and *Col1a1* in response to oxLDL (50 µg/mL for 48 h) was more robust in LDAH-Tg macrophages than in WT controls (Fig. 6E). However, consistent

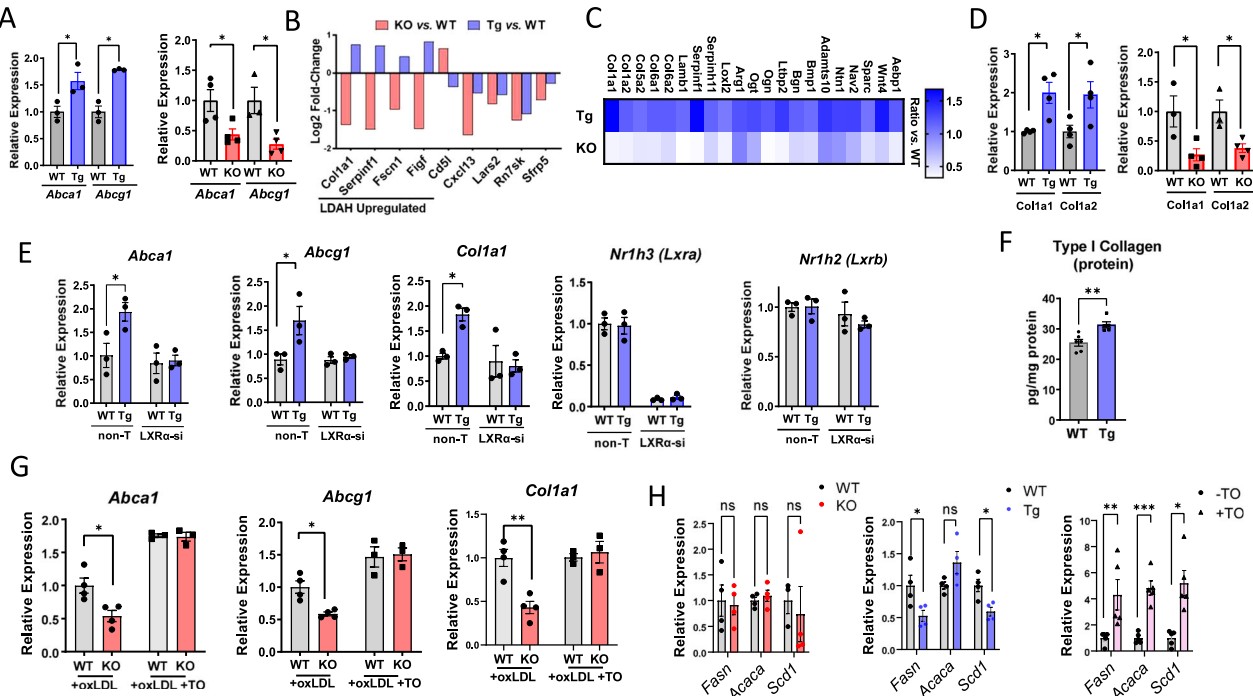

**Fig. 6 | LDAH induces LXR-dependent transcriptional changes. A** qPCR analysis of *Abca1* and *Abcg1* mRNA in PM isolated from LDAH-Tg (Tg, blue bars) and LDAH-KO (KO, red bars) mice treated with oxLDL (50 μg/ml) for 48 h *vs.* PM isolated from their respective WT controls (gray bars) (n = 3-4 independent samples). **B** Common genes identified in RNA-seq analysis of LDAH-KO (red bars) and LDAH-Tg (blue bars) PM *vs.* their WT controls at q-value < 0.05. **C** RNA-seq analyzes identified multiple ECM-related genes induced by LDAH. The upper row in the heat map (Tg) represents the ratio of LDAH-Tg PM *vs.* WT, and the lower row (KO) represents the ratio of LDAH-KO PM *vs.* their WT controls (n = 4). **D** qPCR analysis of Col1a1 and Col1a2 genes in LDAH-Tg (Tg, blue bars) and LDAH-KO (KO, red bars) *vs.* their respective WT control (gray bars) PM treated with oxLDL (50 μg/ml) for 48 h (n = 3–4). **E** WT (gray bars) and LDAH-Tg (Tg, blue bars) BMM were treated with siRNA against LXRα (LXR-si) or with non-target siRNA, followed by oxLDL (50 μg/ml) for 48 h. Expression of *Nr1h3 (Lxra)*, *Abca1*, *Abcg1*, and *Col1a1* was determined by qPCR (n = 3). (**F**) ELISA quantification of pro-collagen I alpha in WT (gray bar) and LDAH-Tg (Tg, blue bar) BMM treated with ox LDL (50 μg/ml) for 48 h (n = 5–6).

**G** LDAH WT (gray bars) and KO (red bars) PM were treated with oxLDL (50 μg/mL) for 48 h and media was replaced by media with or without TO901317 (5 μM) for 6 h. TO rescued the downregulation of *Abca1, Abcg1.* and *Col1a1* seen under LDAH deficiency (n = 4). **H** qPCR analysis of *Scd1, Fasn* and *Acaca* mRNA in LDAH-Tg (Tg, blue bars) and LDAH-KO (KO, red bars) PM treated with oxLDL (50 μg/ml) for 48 h *vs.* their respective WT controls (gray bars) (n = 3-4). In the right panel, WT PM were treated with oxLDL (50 μg/ml) for 48 h and media was replaced by media with (+TO, purple bars with triangles) or without (-TO, gray bars with circular dots) TO901317 (5 μM) for 6 h (n = 5). Comparisons in (**A**), (**D**), (**E**) and (**H**) were performed by two-tailed unpaired *t*-test (normally distributed with equal variances), two-tailed Welch's *t*-test (normally distributed with unequal variances), or two-tailed Mann-Whitney U (not normally distributed). Comparisons in (**F**) and (**G**) were performed by two-way ANOVA followed by Tukey's test. *$p < 0.05$, **$p < 0.01$, ***$p < 0.001$. All bars represent mean ± SEM of independent samples. Source data are provided as a Source Data file.

with an LXR-dependent mechanism, the enhanced expression of *Abca1*, *Abcg1*, and *Col1a1* seen in LDAH-Tg macrophages was hindered under LXRα knockdown (Fig. 6E). Furthermore, treatment with the synthetic LXR agonist TO901317 rescued the downregulation of *Abca1*, *Abcg1* and *Col1a1* in LDAH-KO macrophages (Fig. 6G). Despite the high efficacy of some of the available synthetic LXR agonizts, their development into therapeutics has been hindered by lipogenic side effects[32]. In contrast, natural sterol LXR ligands can avoid lipogenic pitfalls thanks to their ability to coordinately downregulate the processing of sterol-regulatory element-binding protein-1c (SREBP)-1c[32–34]. Our RNA-seq data revealed that some lipogenic enzymes, including stearoyl CoA desaturase-1 (*Scd1*), were downregulated in LDAH-Tg macrophages (Table S5). Thus, we assessed the expression of main lipogenic enzymes, *Scd1*, fatty acid synthase (*Fasn*) and acetyl CoA carboxylase alpha (*Acaca*), and found that, while the expression of these genes was significantly induced upon treatment with TO901317, transgenic LDAH overexpression did not affect (*Acaca*) or even reduced (*Fasn, Scd1*) the expression of these genes (Fig. 6H). Taken together, although these studies do not exclude other potential protective mechanisms, the loss-of-function and rescue experiments suggest that LDAH induces atheroprotective and pro-stabilizing transcriptional changes through an LXR-dependent mechanism, and interventions to enhance the lipolytic release of regulatory sterols might prove suitable to selectively enhance LXR activation.

## LDAH reduces sterol ester accumulation in vivo and promotes an anti-inflammatory and pro-fibrotic foam cell phenotype

To assess whether LDAH's effects on primary macrophages might be reflected in vivo, we first assessed LDAH's distribution among macrophage subsets and its correlation with other genes using the single cell transcriptomic sequencing data from Kim et al. (GEO PRJNA477941), a study that included both total arterial leukocytes and foam cells[35]. As reported, foam cells consisted mainly of populations poor in inflammatory transcripts (Supplementary Figs. S11, S12). Accordingly, the pro-inflammatory cytokine IL-1β that is enriched in pro-inflammatory macrophage clusters, was markedly reduced in foam cells (Supplementary Fig. S11). In contrast, LDAH appeared enriched in the non-inflammatory foam cells, and visually correlated with LXRα and β, ABCA1 and Col1a1 (Supplementary Figs. S11–S13). Furthermore, Pearson's correlation of LDAH with all other expressed genes in total (foamy and non-foamy) arterial CD45+ cells coupled with gene ontology analyzes identified numerous metabolic regulators, including LXRα and many of its targets, as well as genes involved in ECM homeostasis, including *Col1a1* and *Col1a2*, but yielded a low number of inflammatory genes and inflammation-related terms (Supplementary

Table S9 and Supplementary Fig. S14). These in vivo data correlate LDAH with less inflammatory macrophage populations and with fibrotic markers and LXR targets identified in the transcriptomic analyses of primary macrophages.

Next, we investigated LDAH's effects on plaque sterol ester accumulation in situ in lesions of WT and LDAH-Tg male mice using high-resolution matrix-assisted laser desorption/ionization mass spectrometry imaging (MALDI-MSI), a technique that takes into consideration the spatial distribution of the analytes in a tissue section, and therefore allows focusing the analysis on the actual atherosclerotic lesions[36]. As in mice used for atherosclerotic studies (Fig. 2B), at 20 weeks of age the lesions of male mice of both genotypes consisted primarily of Mac3-positive fatty streaks, and the ratio of Mac3 to total lesion area was similar between genotypes (Fig. 7A). In shotgun lipidomics of primary macrophages, LDAH overexpression did not affect the levels of phosphatidylcholines (PC) and other phospholipids that are abundant constituents of cellular membranes (Fig. 4A, B). Consistently, the intensities of total ion count (TIC) normalized images corresponding to main PC species were similar between lesions of WT and Tg mice (Fig. 7B). In contrast, in agreement with the lipidomic data from primary macrophages, the intensities of the images corresponding to the main CE species identified in the analysis were overall lower in lesions of LDAH-Tg mice than in their WT littermates (Fig. 7C, D and Supplementary Table S10). In addition, quantification of imaging data corresponding to compounds with m/z values and predicted molecular formulas that match esters of oxidized sterols also showed lower intensities in lesions of transgenic mice, suggesting that LDAH also impacts a broad range of esterified sterol species in vivo (Fig. 7E and Supplementary Table S10).

For analysis of gene expression, we used laser capture microdissection (LCM) to selectively isolate macrophages from lesions of WT and LDAH-Tg male mice[37,38]. As in primary macrophages, the expression of Abca1, Col1a1, and Col1a2 genes was higher in macrophage/foam cells of $Ldah^{Tg/0}Apoe^{-/-}$ mice than in $Ldah^{0/0}Apoe^{-/-}$ littermates, while expression of the lipogenic genes Fasn and Acaca remained similar between genotypes (Fig. 7F). LXR activation suppresses inflammation, and the role of macrophages in collagen homeostasis is highly dependent on their polarization status[32,39,40]. While classically activated pro-inflammatory macrophages are predominantly collagenolytic, alternative activation into a reparative phenotype has been associated with both increased collagen synthesis and reduced degradation[41]. Expression of arginase 1 (Arg1), a key enzyme in the generation of proline precursors for collagen synthesis and a common marker of alternative macrophage activation whose expression in atherosclerotic plaques was shown to be LXRα-dependent and was upregulated in LDAH-Tg PM in our RNA-seq analyzes, was also significantly higher in LDAH-Tg lesional macrophages (Fig. 7F)[42]. In addition, the expression of the anti-inflammatory cytokine Il10 was significantly induced by ~4-fold, and expression of the pro-inflammatory cytokine Il18 was significantly reduced in LDAH-Tg foam cells (Fig. 7F). There was also a non-statistically significant reduction in the mRNA levels of other pro-inflammatory mediators, including Tnfa, Il1b, Cox2, and two chemokines previously found reduced in response to hypercholesterolemia in vivo, Cxcl9 and Cxcl10 (Fig. 7F)[43]. The expression of two matrix metalloproteinases, Mmp2 and Mmp9, which were previously found downregulated by LXR, was also significantly lower in LDAH-Tg macrophages, whereas the expression of tissue inhibitor of metalloproteinases-1 (Timp1) was upregulated (Fig. 7F)[44,45].

Taken together, the in vivo analysis of atherosclerotic lesions also supports the role of LDAH in the mobilization of a broad range of esterified sterols, including endogenous LXR ligands. Gene expression analyses of foam cells within the atherosclerotic milieu suggest that LDAH promotes a reparative macrophage phenotype, and that parallel to the upregulation of collagen-encoding genes, LDAH tones down inflammation and the macrophage's collagenolytic program. The main effects of LDAH and proposed mechanisms of action in foam cells and atheroma are summarized in Fig. 8.

## Discussion

LDAH is a LD-associated protein with a conserved lipase structure, high affinity for LDs of lipid-laden macrophages, and abundantly expressed by foam cells in atheroma[10]. To investigate LDAH's lipid substrates and roles in foam cell biology and atherogenesis, we have generated LDAH-Tg and LDAH-KO mice. The combined gain- and loss-of-function studies consistently show that LDAH limits atherosclerosis development and fosters stable lesion architectures, as characterized by reduced necrosis and increased deposition of fibrillar collagen. Atherosclerosis studies performed on mice with their bone marrows reconstituted with LDAH-deficient cells and studies under LDAH overexpression driven by a myeloid promoter suggest that LDAH expression in macrophage/foam cells is sufficient to induce these favorable effects.

LDAH's strong affinity for LDs and lipase structure suggests a role in the mobilization of lipid esters from LD compartments. We first investigated how LDAH impacts the foam cell's lipidome using unbiased shotgun lipidomics, which identified sterols as the lipids most consistently and reciprocally impacted by LDAH gain- and loss-of function. This contrasts with a previous report that did not observe differences in cholesterol metabolism between WT and LDAH-deficient macrophages, nor identified any other candidate LDAH substrate[46]. However, in this study analysis of CE mass was limited to densitometric quantification of thin-layer chromatography (TLC) bands, and CE mobilization was assessed using methyl-ß-cyclodextrin as the acceptor, a chemical that strips free cholesterol from the plasma membrane[46]. In our studies, all individual CE species identified in the high-resolution shotgun lipidomics were reduced by LDAH, regardless of the fatty acid esterified to the sterol. Subsequent trafficking experiments using the physiological cholesterol acceptor apoA-I suggest that, as anticipated from a lipase/esterase, the reduction in sterol esters is due to enhanced ester hydrolysis. In addition, high-resolution MALDI-MSI analysis of atheroma also supports a significant role of LDAH in sterol homeostasis in vivo.

While lipoproteins can elicit pro-inflammatory responses, for example through engagement of toll-like receptors (TLRs)[47], a growing number of studies using primary foam cells and single cell transcriptomics of murine and human atheroma indicate that foam cells are not main drivers of inflammation[35,43,48,49]. In addition to high abundance lipids such as cholesterol and triglyceride, foam cell formation leads to the accumulation of a large number of lipid species, including potentially protective metabolites that could counter other pro-inflammatory signals of lipoproteins. Given that many of the lipids that accumulate in foam cells can be stored as esters in cytoplasmic LDs, the trafficking in and out of LD compartments could significantly modulate the effects of bioactive species[2–6]. LDAH was shown to regulate the activity of an anti-cancer macrolide that partitions to LDs by cleaving an ester bond in its side chain, which leads to the release into the cytosol of a more hydrophilic metabolite that kills cancer cells more efficiently[50]. However, whether LDAH also regulates endogenous bioactive metabolites that traffic through LDs is not known. Sterol panels and MALDI-MSI analysis of atheroma suggest a broad impact of LDAH on sterols, including species that are endogenous LXR ligands. Trafficking experiments suggest that increased ester hydrolysis by LDAH might provide a mechanism of access of regulatory sterols to their receptors. While this mechanism may also apply to desmosterol, a cholesterol synthesis intermediate that is also esterified in macrophages and in our lipidomic studies followed a pattern similar to other sterols, further research will be needed to determine whether desmosterol is also affected by negative feedback of sterols on the synthesis pathway[19]. Our studies identified numerous effects of LDAH that are compatible with a scenario of enhanced LXR activation. In

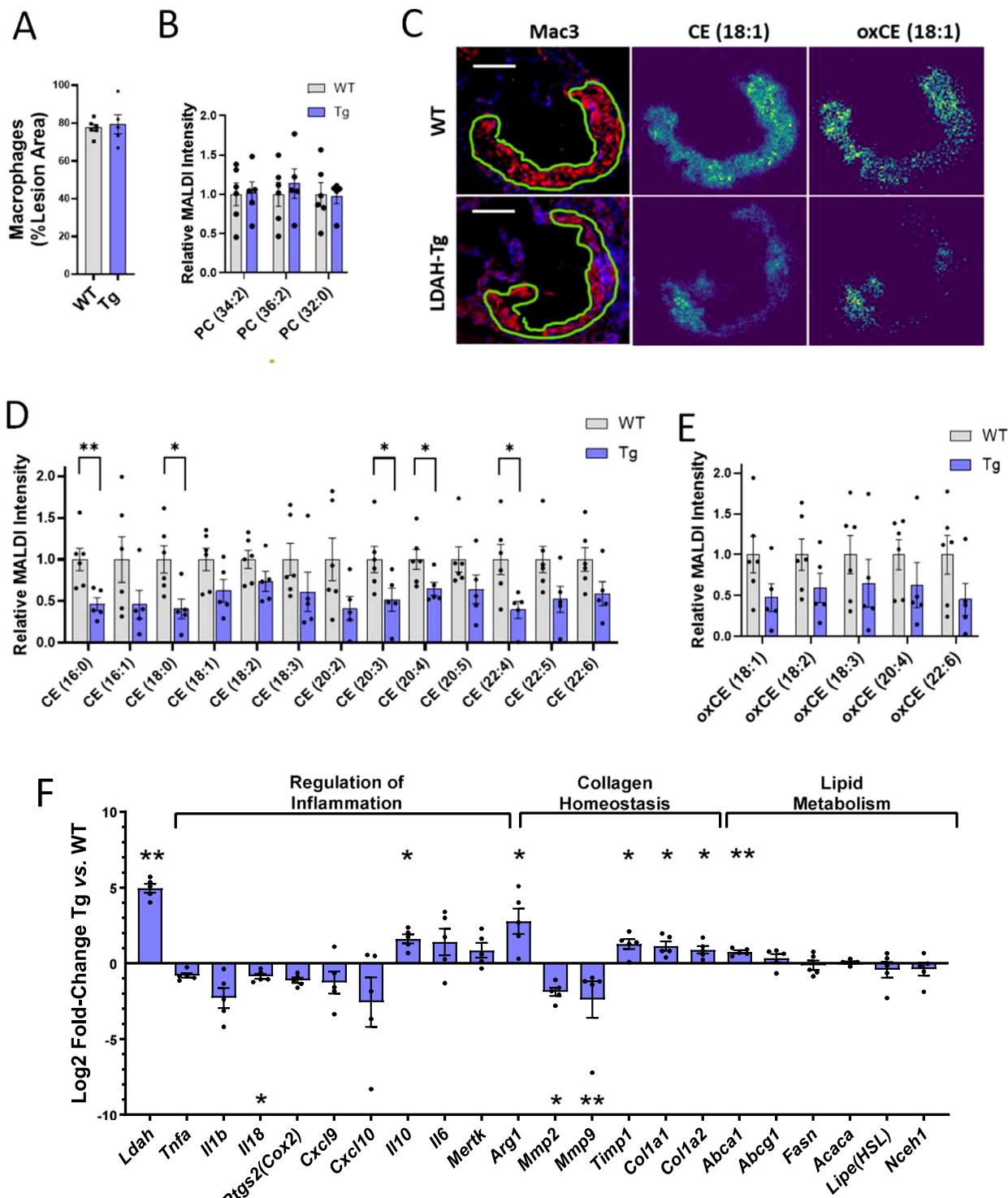

**Fig. 7 | LDAH impacts sterol homeostasis in vivo and leads lesional foam cells to a less inflammatory phenotype with a pro-fibrotic molecular signature. A–E** Atherosclerotic lesions of 20-week-old LDAH-Tg (Tg, blue bars) (*n* = 5) and wild-type (WT, gray bars) (*n* = 6) littermate male mice were analyzed using high-resolution MALDI-MSI. **A** Macrophage content was similar in lesions of mice of both genotypes. **B** Quantification of main structural PC species by lesion area using MALDI-MSI. **C** Exemplary images of consecutive sections stained for Mac3 and MS images representing CE (18:1) and oxidized (ox) CE (18:1). Scale bars: 200 μm. (**D, E**) Quantification of main CE (**D**) and candidate oxidized CE (**E**) species identified in atherosclerotic lesions. The first number in the parenthesis represents the number of carbons of the fatty acid esterified to the sterol, and the number after the colon represents the number of double bonds. **F** Gene expression analyzes. RNA was obtained by LCM from Mac3 positive areas within lesions of LDAH-Tg (Tg) and wild-type (WT) male mice (n = 5 per genotype). The blue bars represent Log2-fold change of Tg over WT. Comparisons were performed by two-tailed unpaired *t*-test (normally distributed with equal variances), two-tailed Welch's *t*-test (normally distributed with unequal variances), or two tailed Mann-Whitney U (not normally distributed). All data in this figure are presented as mean ± SEM of independent samples. *$P < 0.05$, **$P < 0.01$, ***$P < 0.001$. Source data are provided as a Source Data file.

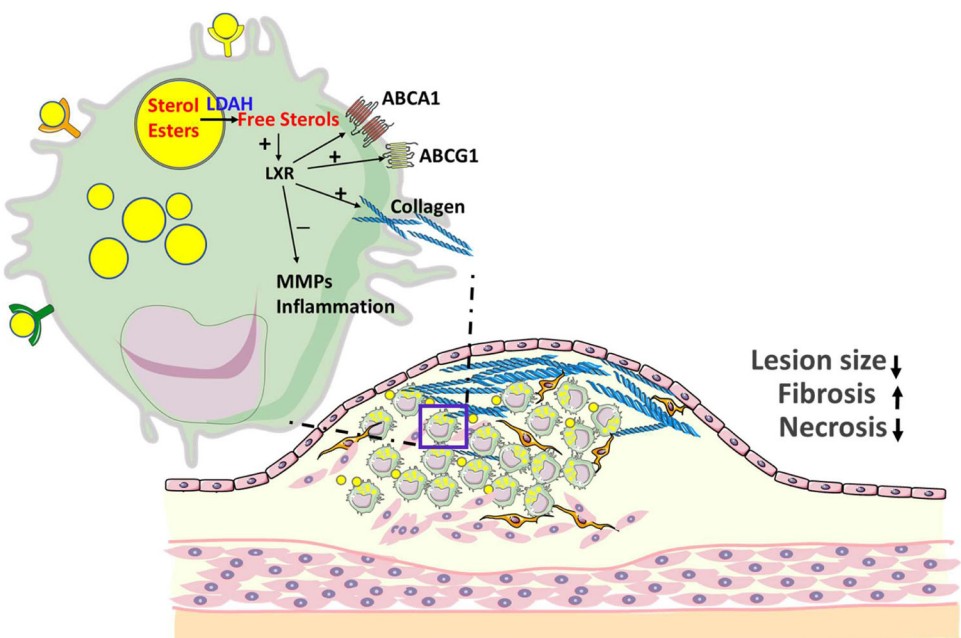

**Fig. 8 | Graphical summary.** LDAH mobilizes stores of esterified sterols that are endogenous LXR ligands and leads to protective modulation of foam cell phenotype, protection against atherosclerosis, and development of more fibrotic and less necrotic lesion architectures. This figure was generated in part using images from Servier Medical Art, under a Creative Commons License CC BY 4.0 (https://creativecommons.org/licenses/by/4.0/).

primary macrophages, LDAH induces the expression of prototypical LXR target genes, and studies under LXRα knockdown and rescue experiments with a synthetic LXR agonist suggest that these effects are LXR-dependent. In atheroma, in addition to protection against lesion development, LDAH leads to more stable lesion phenotypes that are richer in collagen fibers and less necrotic, effects that are also seen upon LXR activation[51–53]. Type I fibrillar collagen is a major contributor to plaque structural integrity, while thin-capped necrotic plaques are more prone to rupture and trigger vascular events[17,54]. Mounting evidence suggests that certain macrophage subsets can directly contribute to constructing their own surrounding matrix, and several studies have linked foam cell formation to alternatively activated macrophage phenotypes with pro-fibrotic features[43,55–58]. Arguably, if the lipid mediators responsible for this phenotype can be tucked away in hydrophobic compartments, their mobilization into the cytoplasm by lipolytic enzymes would exacerbate the phenotype. LXR also suppresses inflammation and supports anti-inflammatory programming of macrophages[32,39,40,42]. Consistently, our analyses show that LDAH plays an anti-inflammatory role, increasing the expression of ARG-1 and IL-10 and reducing the expression of pro-inflammatory cytokines by foam cells. Also consistent with studies linking macrophage activation with their ability to degrade collagen, LDAH reduces the expression of two MMPs that have been shown to parallel inflammation and are downregulated by LXR, and increases expression of the MMP inhibitor TIMP-1[44,45,59,60].

Highly effective synthetic LXR agonizts have been available for decades. However, the clinical use these compounds has been challenged by lipogenic side effects that take place through upregulation of SREBP-1c and downstream genes involved in fatty acid synthesis[61]. A key difference between synthetic agonizts and endogenous sterol agonizts is that, parallel to LXR activation, the sterols coordinately block the activation of SREBP proteins and, as a result, the induction of LXR target genes can be dissociated from lipid synthesis[32,33]. In our studies LDAH did not affect or even reduce the expression of lipogenic genes, suggesting that the lipolytic release of regulatory sterols is devoid of lipogenic pitfalls and, therefore, a metabolic step potentially amenable for therapeutic intervention.

Our lipidomic analyses have consistently identified sterol esters as candidate LDAH substrates. However, foam cells accumulate hundreds of lipid metabolites, and it is possible that some of LDAH's mechanisms of atheroprotection are related to the metabolism of other substrates that we have not been able to identify. Alternatively, in addition to LXR activation, regulatory sterols display other biological functions that could affect atherosclerosis development[24,62,63]. Nevertheless, the fact that a LD enzyme leads to favorable modulation of foam cell phenotype and atheroprotection warrants interest in the study of the mechanisms that regulate the trafficking of bioactive metabolites in and out of LD compartments. Overall, these studies underscore the role of LDs as central storage sites and metabolic hubs of regulatory lipids, and the importance of lipolytic signaling in the pathogenesis of atherosclerosis and potentially other diseases linked to enhanced lipid accumulation.

## Methods
### Mice
All animal studies were conducted in accordance with the NIH Guide for the Care and Use of Laboratory Animals, following protocols approved by the Albany Medical College Institutional Animal Care and Use Committee (protocol number 20-11001). Mice were housed under regular 12/12 light-dark cycles and under regulated temperature of 20–26 °C and a humidity level of 30–70%. Mice were fed standard chow (Prolab® Isopro® RMH 300-5P76, LabDiet) or a WD containing 21% wt/wt milk fat and 0.2% wt/wt cholesterol (Envigo TD.88137). Mice were euthanized by carbon dioxide inhalation or by cervical dislocation under general anesthesia induced by isoflurane inhalation. Both male and female mice were used for this study and analyzed independently. LDAH transgenic (LDAH-Tg, $Ldah^{Tg/0}$) mice in B6D2F1 background were generated at the Rodent Genetic Engineering Laboratory Technologies at the NYU Langone Medical Center. Transgenic LDAH expression is driven by a myeloid-specific promoter that contains the proximal promoter and first intron of the mouse $Csf1r$ gene, the gene that encodes the receptor for macrophage colony-stimulating factor (CSF1), and includes a critical intronic enhancer element (FIRE)[15,16]. Positive founders were identified by PCR and

crossed 7 times with C57BL/6 mice, followed by two additional crosses with *Apoe*$^{-/-}$ mice, also in C57BL/6 background, to generate *Ldah*$^{Tg/0}$*Apoe*$^{-/-}$ mice. *Ldah*$^{Tg/0}$*Apoe*$^{-/-}$ mice were bred with *Apoe*$^{-/-}$ mice to produce *Ldah*$^{Tg/0}$*Apoe*$^{-/-}$ and *Ldah*$^{0/0}$*Apoe*$^{-/-}$ littermates. LDAH-knockout (KO, *Ldah*$^{-/-}$) mice were generated directly in the C57BL/6 background by homologous recombination at the KOMP repository (www.komp.org), using an expression-selection cassette to replace the first two coding exons (exons 2 and 3) and part of exon 4. Heterozygous F$_1$ mice were identified by PCR, and complete knockout in homozygous was confirmed by PCR and western blotting. *Ldah*$^{-/-}$ mice were crossed twice with *Apoe*$^{-/-}$ mice to generate *Ldah*$^{+/-}$*Apoe*$^{-/-}$ mice, which were then used as breeders to produce littermate *Ldah*$^{+/+}$*Apoe*$^{-/-}$ and *Ldah*$^{-/-}$*Apoe*$^{-/-}$ mice. For bone marrow transplant, *ApoE*$^{-/-}$ mice were subjected to lethal irradiation (950 rads in a 3-hour interval split-dose of 475 rads per lift) and their bone marrows were reconstituted with cells from *Ldah*$^{-/-}$*Apoe*$^{-/-}$ or *Ldah*$^{+/+}$*Apoe*$^{-/-}$ donors[64].

## Analysis of atheroma and plasma lipid profiles

After euthanasia, hearts were perfused with phosphate-buffered saline, excised, dissected by a parallel cut under the tip of the atria, and immediately embedded in optimal cutting temperature (O.C.T.) compound and stored frozen at -80 °C. Hearts were cryosectioned (7 µm) through the aortic sinus. Cryosections spanning the entire aortic sinus were stained with oil red O (ORO), and the size of atherosclerotic lesions was measured in 7 representative sections[64,65]. Consecutive sections were stained with Masson's trichrome and H&E and used to quantify lesional collagen and areas of necrosis, respectively. Additional consecutive sections were immunostained with anti-Lamp2/Mac3 (Santa Cruz Biotechnology, SC-19991 dilution 1:200 or SC-19991AF647 dilution 1:100) and with anti-COL1A1 (Sigma-Aldrich, 234167, dilution 1:50). Apoptotic cells were identified using a terminal deoxynucleotidyl transferase dUTP nick end labeling (TUNEL) assay (Roche, 12156792910) following the manufacturer's instructions[66]. The intensities and/or areas of staining were determined using ImageJ (v. 1.53) and AxioVision V.4. (Carl Zeiss Microscopy) software. Lipoproteins in plasma pools of each experimental group were fractionated by fast performance liquid chromatography (FPLC) gel filtration[38].

Total plasma cholesterol and HDL cholesterol were determined using Cholesterol E (Fujifilm Wako Chemicals, 999-02601) and HDL-Cholesterol E (Fujifilm Wako Chemicals, 997-01301). Plasma triglyceride was quantified using Infinity Triglyceride Reagent (Thermo Scientific, TR22421).

## Culture of primary macrophages and siRNA

Peritoneal macrophages (PM) were harvested 5 days after aged 3% thioglycolate injection. For lipidomic and transcriptomic analyzes, after overnight culture in DMEM-10% FBS the media was replaced by DMEM-1% FBS supplemented with Hi-TBAR oxLDL (50 µg/ml, Alfa Aesar, J652618PL) for 48 h. To generate bone marrow-derived macrophages (BMM), bone marrow cells were cultured in RPMI-1640 supplemented with 10% FBS and 15% L929-conditioned media[67]. For studies on LXR downregulation, fully differentiated BMMs were plated at 1 - 2×10$^6$ density on 60 mm dishes. Cells were transfected with siRNA pools using On-TARGETplus Mouse Nr1hr2 siRNA (22260), Nr1hr3 (22259), and Non-targeting Pool siRNA (D-001810-10-20) from Dharmacon. Transfected cells were incubated for 48 h, followed by treatment with oxLDL. The siRNA sequences are listed in Supplementary Data 1.

## Western blotting

Approximately 8.5×10$^5$ to 1×10$^6$ cells cultured in 60 mm culture dishes were rinsed with 1xPBS (pH 7.4) and lysed in RIPA buffer (50 mM Tris-HCl, pH 7.5, 150 mM NaCl, 1% Nonidet P-40, 0.5% sodium deoxycholate, 0.1% SDS, 2 mM EDTA) with protease inhibitor cocktail (Roche). Cells were centrifuged at 20,000 × *g* at 4 °C for 5 min, and

protein concentration in the supernatants was measured using DC™ Protein Assay Kit II (Bio-Rad Laboratories). Approximately 25 µg of protein lysates were resolved by SDS-PAGE and transferred to PVDF membranes. After blocking, membranes were incubated overnight with a custom polyclonal rabbit anti-mouse LDAH antibody generated at Bethyl Laboratories (dilution 1:3000), with anti-ABCA1 (Nobus Biologicals, NB400-105, dilution 1:1000), with anti-beta-Actin (Sigma-Aldrich, A5441, dilution 1:5000) or with anti-GAPDH (Sigma-Aldrich, G9545, dilution 1:10000). Incubations with secondary antibodies were performed at room temperature for one hour.

## Collagen ELISA

Fully differentiated BMMs were plated at 2×10$^6$ density on 60 mm dishes and treated with Hi-TBAR oxLDL (Alfa Aesar, J652618PL, 50 µg/ml) for 48 h. Cell lysates were collected using the buffer provided in the Mouse Pro-Collagen I alpha 1 ELISA Kit (Abcam, ab210579). Pro-collagen in lysates of BMM was quantified following the manufacturer's instructions and normalized to total protein.

## Lipidomics and sterol trafficking

Approximately 4 − 5×10$^6$ PM were subjected to monophasic lipid extraction in methanol:chloroform:water (2:1:0.74, *v:v:v*). Pelleted cellular proteins were subsequently dissolved in 0.1 N NaOH to determine the total amount of protein per sample group using DC™ Protein Assay Kit II (Bio-Rad Laboratories) and used for the normalization of lipids detected. Di-myristoyl phosphatidylcholine (Avanti Polar Lipids, Alabaster, AL, USA) and 19-hydroxycholesterol (Steraloids, Newport, RI, USA) were added to samples during lipid extraction as internal standards for relative quantitation of lipids and sterols. Dried lipid extracts were desalted by washing and resuspended using 200 µL/mg protein in a solution of methanol containing 0.01% butylated hydroxytoluene. The samples were stored under a blanket of nitrogen at −80 °C until further analysis. For shotgun lipidomics, lipid extracts were diluted into isopropanol: methanol (2:1,*v:v*) containing 20 mM ammonium formate and analyzed by flow injection high resolution/accurate MS and tandem MS. Lipid species were identified using the Lipid Mass Spectrum Analysis (LIMSA) v.1.0 software linear fit algorithm, in conjunction with a user-defined database of hypothetical lipid compounds for automated peak finding and correction of $^{13}$C isotope effects. Relative quantification of lipid abundance between samples was performed by normalization of target lipid ion peak areas to the di-myristoyl phosphatidylcholine internal standard[68]. Sterols and oxysterols were analyzed by high resolution/accurate mass LC-MS using a Shimadzu Prominence HPLC coupled to a Thermo Scientific LTQ-Orbitrap Velos mass spectrometer[3]. Chromatographic peak alignment, compound identification, and relative quantitation against the 19-hydroxycholesterol internal standard were performed using MAVEN software 3.6.1. For analysis of cholesterol and 25-hydroxycholesterol (25-HC) esterification and hydrolysis, PM was cultured in DMEM-0.2% BSA containing acetylated low-density lipoprotein (acLDL, 50 µg/ml, Alfa Aesar, J650298PL) and 25-HC (1.5 µg/ml, Sigma-Aldrich, H1015) for 24 h. Cells were washed, pulsed for 19 h with oleate (0.2 mM) labeled with [9,10 (n) $^3$H]-oleic acid (Perkin Elmer) in DMEM-0.4% BSA, washed, and chased in media containing apoA-I (10 µg/ml, Alfa Aesar, J64506MCR) and an ACAT1 inhibitor (10 µg/ml, Sandoz 58-035, Sigma-Aldrich, S9318) for 6 and 22 h. Cellular lipids were extracted with hexane: isopropanol (3:2, *v:v*). Protein pellets were dissolved in 0.2 N NaOH, neutralized with HCl, and total protein content was quantified using DC™ Protein Assay Kit II (Bio-Rad Laboratories). Cellular lipids were resolved by TLC using hexane-diethyl ether-glacial acetic acid (75:35:1) and the amounts of $^3$H-labeled CE and 25-HC ester (25-HCE) were determined by scintillation counting and normalized to protein[64]. To identify the 25-HCE and CE bands, highly pure 25-HC oleate was purchased from Steraloids and combined with TLC standard reference mixtures (Nu-Check Prep, TLC-18-6C and TLC-

18-4A). The amounts of $^3$H-labeled CE and 25-HCE were determined by scintillation counting and normalized to protein.

## High-resolution MALDI MS imaging

Aortic sinuses embedded in 5% gelatin (VWR) were cryosectioned (10 μm) and tissue slices were collected on Superfrost Plus Microscope Slides (Fisher Scientific, 12-550-15)[36]. The samples were measured with an atmospheric-pressure MALDI imaging ion source (AP-SMALDI[5] AF, TransMIT GmbH, Giessen, Germany), coupled to an orbital trapping mass spectrometer (Q Exactive HF, Thermo Fisher Scientific (Bremen) GmbH, Germany)[69]. Measurements were performed in positive-ion mode using a mass range of $m/z$ 300-1200, a mass resolution of R = 240,000 @ $m/z$ 200, and a spatial resolution of 7 μm. Internal mass calibration was achieved using the lock-mass feature of the orbital trapping mass spectrometer resulting in a mass accuracy of <3 ppm. For data analysis, raw data of all measurements were stitched together and converted to imzML format. Annotation of lipid signals ([M + H]$^+$, [M+Na]$^+$, [M + K]$^+$) was performed by Metaspace, a fully automated metabolite annotation platform for imaging data, using the databases of HMDB-v4, LipidMaps -2017-12-12 and SwissLipids-2018-02-02 and a ± 3 ppm m/z window[70]. A custom database was used for oxidized CE lipids, which are not listed in the standard databases. Regions of interest (ROI) comprising lesion area were defined for each sample. Total ion count (TIC) normalized images of annotated lipids (+/- 3 ppm) containing the ROIs of all samples were exported from MIRION software as.csv format. In Matlab, the maximum intensity of the images was set to the 99th percentile, and the intensity sum of each lipid signal in the ROI was divided by the pixel number of the ROI. Calculated values were normalized to the area of macrophages determined by Mac-3 staining.

## Analysis of gene expression

RNA was extracted with TRIzol (Invitrogen) and purified with Absolutely RNA Miniprep Kit (Agilent). For PolyA-enriched RNA sequencing, RNA quality was assessed using an Agilent Bioanalyzer and quantified using a Qubit Fluorometer by the Functional Genomics Core at the Roy J. Carver Biotechnology Center, UIUC. Hi-Seq libraries were prepared, and 100 bp paired-end Illumina sequencing was performed on a HiSeq 4000 at the High Throughput Sequencing and Genotyping Unit, UIUC. RNAseq reads were processed for quality and read length filters using Trimmomatic (version 0.38). RNAseq reads were further aligned to the mouse genome (mm10) using STAR (version 2.4.2a). Gene expression levels were determined using count and differential expression[71]. Differences in gene expression between experimental groups (LDAH-KO *vs.* WT littermate controls and LDAH-Tg *vs.* WT littermate controls) were considered significant when the false discovery rate adjusted p-value (q value) was < 0.05. Gene ontology and pathway analyses were performed using DAVID knowledgebase v2023q4[72,73]. To identify potential LXR target genes among the genes regulated by LDAH, we compared the Ldah-Tg DEG list with a list of candidate LXR target genes previously identified in macrophages using genome-wide ChIP-seq and transcript profiling[26,27]. For in vivo analysis of gene expression in foam cells, RNA from macrophage-rich areas within atheroma was isolated by laser capture microdissection (LCM) and amplified by in vitro transcription (IVT)[10,37,38]. Relative gene expression levels were determined from threshold cycle (Ct) values normalized to cyclophilin A. Primer sequences are listed in Supplementary Data 1.

## Analysis of single cell sequencing datasets

For reanalysis of the published Kim et al. dataset of total LDLR KO leukocytes and ApoE KO foam cells, the corresponding single-cell data sets (GEO PRJNA477941) were downloaded[35]. These Cell Ranger filtered genes, by barcode and expression matrices, were used as analysis inputs into the Partek Flow version11.0.24.0624 software package. The two data sets were combined at the beginning and split only after the clustering to capture all the variations and commonalities in the two models. Cells with a mitochondrial fraction >5% of the fraction of genes expressed were filtered out. Noise reduction was performed by excluding features wherein the value was 0 in at least 99.95% of cells. Normalization was performed on the remaining 57.33% features by expressing all the values as counts per million in each cell, and then the values were Log2 transformed. The dimensional reduction was performed by PCA using the top 2000 features with the highest variance, and the top 10 PCA components were used based on the elbow method for a K-means clustering using Euclidean distance and 1000 iterations with 11 clusters. This was arrived at after testing to arrive at a resolution of the inflammatory subsets and guided by published data from Kim et al. UMAP plots were then constructed for visualization. The cell clusters were annotated based on the biomarker list generated for these 11 clusters using Cell ACT annotation tool (http://xteam.xbio.top/ACT/)[74]. A Pearson correlation analysis was done for all expressed genes for LDAH and those with a FDR of <10e-7 were used for functional enrichment analysis using DAVID knowledgebase v2023q4.

## Statistics

The number of biological replicates is stated in the figure legends. All data are presented as the mean and standard error of the mean (SEM). Data that followed a normal distribution and had equal variances were compared using unpaired two-tailed *t*-test or by two-tailed Welch's *t*-test if variances were unequal, and data that did not follow a normal distribution were analyzed using two-tailed Mann-Whitney U. Multiple comparisons were performed using 2-way ANOVA, followed by Tukey's test.

## Reporting summary

Further information on research design is available in the Nature Portfolio Reporting Summary linked to this article.

# Data availability

The RNA sequencing data generated in this study have been deposited in Gene Expression Omnibus with accession number GSE157947 at. The shotgun lipidomic raw data have been deposited in Metabolomics Workbench[75] under study IDs ST003317 and ST003316 at https://doi.org/10.21228/M8TF97 and https://doi.org/10.21228/M8Z823. The mouse scRNA-seq data reanalyzed in this study were downloaded from Gene Expression Omnibus under accession code GSE116271 at[35]. The sequences of primers and siRNAs used in this study are available in Supplementary Data 1. All other data supporting the findings in this study are available in Supplementary Information/Source Data files. Additional information, resources, and reagents are available from the corresponding authors upon request. Source data are provided with this paper.

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

## Acknowledgements

We thank Dr. David A. Hume for providing the promoter and enhancer of the *Csf1r* gene for the transgenic construct. The knockout mouse strain used for this research project was generated from targeted ES Cells for 1110057K04Rik obtained from the KOMP Repository www.komp.org, a NCRR-NIH supported mouse strain repository (U42-RR024244). ES cells from which this mouse was generated were created by Velocigene from funds provided by the trans-NIH Knock-Out Mouse Project (KOMP) (Grant # 5U01U01HG004085). Product inquiries can be emailed to service@komp.org. We thank the Roy J. Carver sequencing and genotyping core facilities at the University of Illinois, Urbana-Champaign. Figure 8 was generated in part using images from Servier Medical Art, under a Creative Commons License CC BY 4.0 (https://creativecommons.org/licenses/by/4.0/). This research was funded by grants from the American Heart Association (18TPA34230103 to A. Paul and 971847 to Y.-H. Goo); A. Paul was supported in part by the National Institutes of Health (R01DK128972); A. Kalsotra was supported by the National Institutes of Health (R01HL126845, R01AA010154), by the Muscular Dystrophy Association (MDA514335), and by a Chan-Zuckerberg Biohub Chicago Investigator Award; S. Bangru was supported by the NIH Tissue microenvironment training program (T32-EB019944) and Scott Dissertation fellowship from UIUC graduate college. V. Yechoor was supported in part by the National Institutes of Health (R01DK130499) and by the US Department of Veterans Affairs (I01BX002678). B. Spengler received financial support by the Deutsche Forschungsgemeinschaft (Sp314/13-1).

## Author contributions

Y.-H.G. and A.P. conceived the study and wrote the manuscript with input from all authors. Y.-H.G., A.P., J.P.A., and F.D.C. conducted atherosclerosis studies, sterol trafficking experiments, and qPCR analyses. P.K.S. conducted lipoprotein analysis. T.A.L. contributed to lipidomic studies. S.B. and A.K. performed the RNA-seq analyses. V.K.Y. analyzed scRNAseq data and provided critical input for data presentation and manuscript writing; P.B. and S.S. performed the MALDI experiments and analyzed the data, and A.H. W. and S.S. provided critical input for the design of these experiments and data presentation.

## Competing interests

B.S. is a consultant of TransMIT GmbH, Giessen, Germany. The other authors declare no competing interests.
