## [Peer Review File · Nature Communications]

Lipid droplet-associated hydrolase mobilizes stores of liver X receptor sterol ligands and protects against atherosclerosisREVIEWER COMMENTS

Reviewer #1 (Remarks to the Author):

The manuscript by Goo et al. investigates the role of lipid droplet-associated hydrolase (LDAH) in atherosclerosis via LXR alpha-activating ligands. Using both LDAH overexpressing macrophage-specific transgenic mice and LDAH-deficient mice, they discern the lipid types influenced by LDAH, its effect on atherosclerosis, and LXR-dependent transcriptional activation. The research is well-conducted, with a key finding being that LDAH is atheroprotective, and its overexpression and inhibition are linked to specific lipid species, including oxysterols. However, the conclusion that the overexpression of LDAH, through oxysterol products and LXR alpha activation, is the driving force behind these protective effects, is less compelling.

A notable observation is the reduction in LXR target ABCA1 expression (Fig 6 E) following the siRNA knockdown of LXR alpha. This is perplexing, especially since LXR beta's presence should sustain oxysterol-driven expression of ABCA1.

Indeed, existing literature suggests that the absence of LXR alpha doesn't affect the oxysterol-driven expression of ABCA1 from peritoneal macrophages (see Repa et al., *Science*, 289 (2000), Figure 6A). It's challenging to understand how the reduction of LXR alpha through siRNA affects ABCA1 expression when LXR alpha's activation is attributed to oxysterols produced by LDAH. The authors need to elucidate why solely losing LXR alpha leads to diminished ABCA1 expression. Does LXR beta expression remain unchanged following LXR alpha siRNA treatment? Additionally, the specifics surrounding the siRNA experiment, including the specificity of the LXR alpha siRNA, should be elaborated upon.

Reviewer #2 (Remarks to the Author):

This study aims to assess the role of LDAH in macrophages. The authors demonstrate a protective effect of LDAH on the development of atherosclerosis, particularly in the formation of more stable lesions enriched with collagen. The study is well-written, and the methods employed, including over-expression and inactivation of LDAH, targeted lipidomic approaches, and MALDI are elegant. The findings compellingly support LDAH's protective role in atherosclerosis development and in forming more stable lesions. They also highlight LDAH's capability in mobilizing cholesterol and oxysterol esters within macrophages. On this latter point, the results confirm those of a previous study from the same group.

The proposed mechanisms are less convincing, and additional experiments would be beneficial in supporting them:

1) Numerous studies have been published on the analysis of immune cells present within atheroma plaques, especially in mice, using single-cell RNA-seq approaches (cochain et al., *circulation Research*. 2018;122:1661-1674 kim et al. *circulation Research*. 2018;123:1127-1142....). The sequencing data from these studies are available in public databases and could readily be used to characterize the macrophage populations expressing LDAH and to correlate LDAH expression with the markers proposed by the authors (LXR targets, fibrosis markers).

2) Lipidomic analysis shows alterations in oxidized cholesterol derivatives, which are natural ligands for LXR. However, it has been suggested that products of cholesterol biosynthesis pathway, particularly desmosterol, might be the most significant agonists. Enhancing lipidomic analyses with desmosterol examination would be insightful.

3) Transcriptomic profiles don't truly reflect the activation of LXR pathways, except for the induction of ABCA1 and ABCG1. It is unclear to the reviewer whether ABCG1 and ABCA1 were found to be significantly altered by either overexpression or underexpression of LDAH in the RNAseq analyses or just by QPCR. This point should be clarified, and any discrepancies should be discussed. The analysis of other LXR target genes, aside from those from the lipogenesis pathway, should be carried out: ApoE, PLTP, LPCAT3, SCD2, to name a few, to confirm the activation of this pathway.

4) The direct production of collagen by macrophages is a novel concept that warrants further substantiation. Are the inductions observed at the mRNA level indicative of collagen production, or are they due to very low basal mRNA levels? This induction is crucial in supporting the authors' proposed concepts and should be confirmed by western blot.

Reviewer #3 (Remarks to the Author):

The interesting manuscript by Paul and colleagues demonstrates that LDAH modulates macrophage activation and protects against atherosclerosis through lipolytic mobilization of endogenous LXR sterol ligands. The authors used LDAH transgenic and knockout mouse models to investigate the role of LDAH in foam cell formation and atherogenesis and identified sterol esters as potential substrates of LDAH using lipidomic analyses of primary macrophages and atherosclerotic lesions. In contrast to the most potent synthetic LXR agonists, the induction of LXR target genes by LDAH was not paralleled by induction of lipogenic genes, which previously argued against the clinical use of LXR agonists. Most of the results support the conclusions and claims, only the quantification of macrophage and collagen content in the plaques needs further clarity.

Major concerns:

Figure 2A, B: It is unclear how exactly the distinct plaque components (e.g. macrophages, necrotic core, and collagen) were quantified as their relative quantification of the respective areas (macrophage + necrotic core + collagen) sum to > 100%. Mac3 staining, especially in Fig. 2B, appears to be unspecific as no cells (nuclei) are visible. Please provide a negative staining control. The same is true for Figure 3 F,G, where macrophage plus collagen areas sum to > 100%. In the section from Tg mice (Fig. 3G), the yellow line that was supposed to delineate collagen and serve for its quantification also includes macrophages and necrotic core. Were these parts also included to determine collagen area? These issues need to be addressed and clarified.

Figure 2: The title states less necrotic core, but this was not determined in males. However, it should be included because it was also indicated in the graphical summary.

Figure 1B-E and throughout the manuscript when Apo^{-/-} are compared to Ldah-tg/ApoE^{-/-} mice: The authors are urged to change the misleading labeling WT and Tg in all figures. In addition, the figure legend to Fig. 1 needs to indicate that Apo^{-/-} and Ldah-tg/ApoE^{-/-} mice are being compared.

Minor concerns:

Introduction: Ref 2 might be deleted because the definition of foam cells is not the main message of this publication and since foam cells in general do not require citation.

Usually, *** indicates $p < 0.001$. Why did the authors choose *** $p < 0.005$?

Please indicate in the figure legends in which subfigures the t-test or Mann-Whitney U test was applied, respectively.

Figure 3F,G: Figure legends are swapped.

Figure S2: Why did the authors perform a two-tailed t-test? Welch correction seems more appropriate because the SDs are different.

We thank the reviewers for taking the time to review this manuscript and for their insightful input to our work. We appreciate that the reviewers think the manuscript is interesting and well written, methods are elegant, and the research is well conducted. However, the reviewers also raised several concerns and provided valuable suggestions, which we have addressed in the revised manuscript.

We trust that this is a constructive and data-driven revision that significantly improves the original submission, and the novel conceptual and mechanistic information in this manuscript will be considered of interest to a broad readership of Nature Communications.

The following is a point-by-point response to the reviewers' comments:

Reviewer #1 (Remarks to the Author):

The manuscript by Goo et al. investigates the role of lipid droplet-associated hydrolase (LDAH) in atherosclerosis via LXRA α -activating ligands. Using both LDAH overexpressing macrophage-specific transgenic mice and LDAH-deficient mice, they discern the lipid types influenced by LDAH, its effect on atherosclerosis, and LXR-dependent transcriptional activation. The research is well-conducted, with a key finding being that LDAH is atheroprotective, and its overexpression and inhibition are linked to specific lipid species, including oxysterols. However, the conclusion that the overexpression of LDAH, through oxysterol products and LXR α activation, is the driving force behind these protective effects, is less compelling.

A notable observation is the reduction in LXR target ABCA1 expression (Fig 6 E) following the siRNA knockdown of LXR α . This is perplexing, especially since LXR β 's presence should sustain oxysterol-driven expression of ABCA1.

Indeed, existing literature suggests that the absence of LXR α doesn't affect the oxysterol-driven expression of ABCA1 from peritoneal macrophages (see Repa et al., *Science*, 289 (2000), Figure 6A). It's challenging to understand how the reduction of LXR α through siRNA affects ABCA1 expression when LXR α 's activation is attributed to oxysterols produced by LDAH. The authors need to elucidate why solely losing LXR α leads to diminished ABCA1 expression. Does LXR β expression remain unchanged following LXR α siRNA treatment? Additionally, the specifics surrounding the siRNA experiment, including the specificity of the LXR α siRNA, should be elaborated upon.

The reviewer has raised a very important point, as some studies such as the pioneering work by Repa *et al.* suggest that LXR α deficiency is not sufficient to inhibit ABCA1 expression due to the redundant function of LXR β .

To address this, first we checked the expression of LXR β in LXR α siRNA transfected cells to assess the specificity of the LXR α siRNA pool used for these experiments. LXR α siRNA did not change LXR β expression (new panel added to Figure 6E). We also tested the effects of both LXR α and LXR β knockdown on ABCA1 expression in bone marrow macrophages (BMM) treated with oxLDL, and consistently found that LXR α knockdown (KD) decreases ABCA1 expression but LXR β KD has no effect (new Fig. S10). To confirm that this is not due to an issue with our experiments, we performed KD experiments in BMM treated with 22-R-OH at 10 μ M as reported by Repa *et al* in their Science paper. As reported, we also found that single KD of LXR α or LXR β did not have significant impacts on ABCA1 expression. These results suggest that the response to a single potent oxysterol (22R-OH) is different from the more complex oxLDL loading system we used.

In cultured macrophages, papers that reported lack of LXR β compensation for LXR α deficiency seem to use lower doses of LXR synthetic ligand (e.g. 1 μ M of T0 compound, PMID 23313547), or lower doses of 22R-OH or other oxysterols (e.g. 3 μ M in PMID 17657314) than the used by Repa *et al* (10 μ M of 22R-OH or T0). Despite some differences in experimental conditions, *in vivo* studies where LXR activation is from natural sterol ligands also suggest different responses to endogenous and synthetic ligands. Two independent laboratories that performed atherosclerosis studies using *Ldlr* KO mice (PMID: 203888921) and *ApoE* KO mice (PMID: 17657314) found that loss of LXR α is sufficient to exacerbate atherosclerosis. However, interventions with synthetic LXR ligands reduced atherosclerosis in both *Lxra* KO/*Ldlr* KO and *Lxra* KO/*ApoE* KO models, suggesting that the level of activation achieved by LXR β and endogenous ligands is unable to compensate for LXR α deficiency, but the highly effective synthetic agonists are able to compensate for LXR α deficiency through LXR β activation.

Based on the comments by the reviewer, to give a more nuanced context to the readers, we have included a more elaborate rationale for the LXR interventions, and we have expanded the information related to siRNA treatments in the methods sections. As mentioned above, based on the reviewer's comments we also include additional supporting data in Fig. 6 and Fig. S10.

New panel in Fig. 6E.

New Fig. S10.

Reviewer #2 (Remarks to the Author):

This study aims to assess the role of LDAH in macrophages. The authors demonstrate a protective effect of LDAH on the development of atherosclerosis, particularly in the formation of more stable lesions enriched with collagen. The study is well-written, and the methods employed, including over-expression and inactivation of LDAH, targeted lipidomic approaches, and MALDI are elegant. The findings compellingly support LDAH's protective role in atherosclerosis development and in forming more stable lesions. They also highlight LDAH's capability in mobilizing cholesterol and oxysterol esters within macrophages. On this latter point, the results confirm those of a previous study from the same group.

The proposed mechanisms are less convincing, and additional experiments would be beneficial in supporting them:

1) Numerous studies have been published on the analysis of immune cells present within atheroma plaques, especially in mice, using single-cell RNA-seq approaches (cochain et al., *circulation Research*. 2018;122:1661–1674 kim et al. *circulation Research*. 2018;123:1127–1142....). The sequencing data from these studies are available in public databases and could readily be used to characterize the macrophage populations expressing LDAH and to correlate LDAH expression with the markers proposed by the authors (LXR targets, fibrosis markers).

This is a great suggestion. To find correlations between Ldah, LXRs and its targets, and fibrosis markers, first we performed t-SNE analyses on the single cell sequencing data from the study by Kim *et al.* (PMID: 30359200, GEOPRJNA477941), as this study included scRNA-seq data of total arterial leukocytes (CD45+) population of *Ldlr*^{-/-} mice and foamy macrophages of *Apoe*^{-/-} mice (New Figures S11-S14, and new Table S9). As reported, IL-1 β was higher in inflammatory clusters. However, its expression was very significantly reduced in foamy macrophages. In contrast, LDAH appears enriched in non-inflammatory foam cells.

In foam cells, t-SNE plots for LDAH, LXR α , LXR β , ABCA1 and Col1a1 show similar expression patterns. Although as anticipated Col1a1 was intense in cells identified as SMCs, it was readily detected in foamy

macrophage populations. The patterns of these genes contrast sharply with the low IL-1 β expression in foam cells (Fig. S13 and see below).

To unbiasedly correlate LDAH with all other expressed genes, we also performed a Pearson correlation analysis in total (foamy and non-foamy) arterial CD45+ cells, coupled with gene ontology analyses of the correlation list. In the supplemental material we have included a list of ~1000 genes with highest correlation with LDAH (FDR <10e-7) (Table S9), as well as a new figure with the results of the GO analysis (Fig. S14 and see below). The correlation list includes numerous genes involved in lipid and cholesterol metabolism and other metabolic processes, including Nr1h3/LXR α and many of its targets, as well as main ECM-related genes that we have identified in our studies such as Col1a1 and Col1a2. In contrast, LDAH's correlation with inflammatory markers was remarkably poor. These correlations, or lack thereof, are reflected in the GO analyses shown in Fig. S14 (also below).

Fig. S14. Gene ontology (GO) and pathway analysis of genes that correlate with LDAH at a false discovery rate (FDR) <e-7 in the single cell RNA sequencing of aortic CD45+ cells and foam cells from mouse atherosclerotic aortas in GSE116240. Panels show the top 20 most significant GO Biological Process (BP) (A) and KEGG pathways (B) identified using the DAVID Bioinformatics platform. The analyses returned multiple metabolism and fibrosis-related terms, in the absence of inflammation-related terms.

2) Lipidomic analysis shows alterations in oxidized cholesterol derivatives, which are natural ligands for LXR. However, it has been suggested that products of cholesterol biosynthesis pathway, particularly desmosterol, might be the most significant agonists. Enhancing lipidomic analyses with desmosterol examination would be insightful.

We have added new data on desmosterol to the targeted sterol panels in Fig. 5 (see below). We have also added references that show that both endogenous and exogenous desmosterol are ACAT substrates (PMID: 7119575; PMID: 2592557). The pattern for desmosterol is very similar to other sterols, which is consistent with a broad impact of LDAH on sterol metabolites, and with the data shown in Fig. 5G, wherein the Tg LDAH overexpressing PM show an increase in hydrolysis of the sterol esters reflected by a decrease in the measured total amount of sterols. While a scenario of increased ester hydrolysis can provide an underlying mechanism of access of regulatory sterols to LXRs, and this could also apply to esterified desmosterol, in the case of desmosterol further research is needed to determine whether the reduced levels primarily owe to changes in trafficking and subcellular compartmentalization, or to negative feedback on the synthesis pathway. These possibilities have been discussed.

3) Transcriptomic profiles don't truly reflect the activation of LXR pathways, except for the induction of ABCA1 and ABCG1. It is unclear to the reviewer whether ABCG1 and ABCA1 were found to be significantly altered by either overexpression or underexpression of LDAH in the RNAseq analyses or just by QPCR. This point should be clarified, and any discrepancies should be discussed. The analysis of other LXR target genes, aside from those from the lipogenesis pathway, should be carried out: ApoE, PLTP, LPCAT3, SCD2, to name a few, to confirm the activation of this pathway.

This is a great suggestion. To identify LXR targets in our RNA-seq outputs, we generated a list of genes based on published genome-wide studies (PMID: 30602495; 22292898). There were 98 overlapping genes of different functions between the significant ($q < 0.05$) changes in LDAH-Tg and the candidate LXR targets, supporting that elevated levels of LDAH regulates gene expression through the LXR pathway (Table S8). Our RNA-seq studies also identified other genes whose expression is regulated by LXR and might contribute to some of LDAH's main effects in atheroma. For example, in macrophages LXR α was shown to enhance Arg1 through indirect mechanisms (PMID: 21757649), and Arg1 was upregulated in the RNA-seq analysis of LDAH-Tg macrophages ($q < 0.05$) and in lesional foam cells of transgenic mice. Similarly, Mmp-2 and Mmp-9, which are reduced by LXR (PMID: 22634718; PMID: 12531895) are also reduced *in vivo* in LDAH-Tg foam cells. In the RNA-seq analysis of primary macrophages MMP-9 was also downregulated in LDAH-Tg macrophages ($p = 0.0074$; $q = 0.08$), very close to the $q < 0.05$ FDR cutoff.

Initially, ABCA1 and ABCG1 were found significantly altered by targeted qPCR analyses following the observation that LDAH mobilizes esters of regulatory sterols that are natural LXR ligands, as these genes are prototypical LXR targets. We have tried to clarify this point in the manuscript. By qPCR we have found both genes upregulated by LDAH in peritoneal macrophages, in bone marrow-derived macrophages, and in foam cells in atheroma. The data from cell culture experiments in this manuscript are representative of multiple experiments with consistent results. To further support that ABCA1 is induced in transgenic macrophages, we are also including western blot data (New Fig S9).

However, ABCA1 and ABCG1 were not among the significant genes in the RNA-seq analysis. In LDAH-KO macrophages ABCA1 expression was reduced by ~30%, $p=0.066$, which is similar to the qPCR

results in KO macrophages. In LDAH-Tg macrophages, ABCA1 was close to controls. However, a qPCR analysis of the same RNA samples used for RNA-seq also found *Abca1* and *Abcg1* increased (see figure). We are not sure why the RNA-seq experiment failed to detect increased ABCA1 in transgenic macrophages. Studies have found that, depending on the analysis workflow, there might be a 15-20% of non-concordant results between RNA-seq and qPCR

(PMID: 33665610). Some of the reasons could include factors such as transcript length bias, sequencing depth, amplification efficiency, etc. that may lead to discrepancies between qPCR and RNA-seq data for any specific gene, despite mitigating measures adopted, including using TPMs and DEseq2 corrections.

With respect to other genes in lipid metabolism, *Scd1* and *Scd2* were lower in transgenic macrophages at a $FDR < 0.05$, but *ApoE*, *Pltp* and *Lpcat3* did not change. Many of the studies on LXR targets have been performed using synthetic ligands, and the effects of more complex systems, namely lipoproteins, remain much less understood. Studies with desmosterol have shown much weaker induction of some of these genes than the achieved with a synthetic agonist, e.g. *Pltp* and *Lpcat3* in Fig. 6C of PNAS PMID: 29632203 are robustly induced by T0 compound, but much less in response to desmosterol. These studies also suggest that, in addition to differences in expression intensity, natural and synthetic ligands can lead to different or even opposing expression patterns of certain genes, particularly lipogenic genes that are induced by SREBP-1c (PMID: 29632203). Endogenous sterols (but not synthetic LXR ligands) bind to INSIG or to SCAP to prevent the SREBP-1c transport from ER to golgi necessary for subsequent transcription of lipogenic genes such as SCDs (PMID: 18974038). This may allow sterol ligands to avoid the lipogenic pitfalls that have prevented the clinical use of the otherwise exceptionally effective synthetic LXR ligands (PMID: 29712848). In our RNA-seq, both *Scd1* and *2* were significantly reduced at $q < 0.05$ in transgenic macrophages, an effect that aligns with the studies comparing desmosterol (certain lipogenic

genes downregulated) with T09 (lipogenic genes upregulated)(PMID: 29632203). We have included additional data on Scd1 in Fig. 6 to complement the data we already had on other lipogenic genes (*Fasn*, *Acaca*).

4) The direct production of collagen by macrophages is a novel concept that warrants further substantiation. Are the inductions observed at the mRNA level indicative of collagen production, or are they due to very low basal mRNA levels? This induction is crucial in supporting the authors' proposed concepts and should be confirmed by western blot.

A growing numbers of studies support that macrophages can contribute to generate their own surrounding matrix, and some studies link lipid metabolism, and potentially LXR activation, with induction of ECM-related genes (e.g.: PMID: 26197235; PMID: 29765329; PMID: 32001677; PMID: 26697355). Studies on atherosclerosis development also link LXR activation with increased plaque fibrosis (PMID: 29394501; PMID: 15539622; PMID: 35477277).

Per the reviewer's previous comment, in the Kim *et al.* dataset we found that Col1a1 is enriched in foamy macrophages, and LDAH expression in the scRNA-seq dataset correlates with collagen genes. Also related to a previous comment by this reviewer, in their 2012 study on desmosterol published in Cell, Spann *et al.* also found genes related to ECM organization enriched in macrophages in response to western diet (see PMID: 23021221, Fig 1D).

In our *in vivo* studies LDAH correlates with intralésional collagen. We show this by trichrome staining and by staining with a type I collagen antibody. Given that type I collagen fibers protect against plaque rupture, this effect might be very relevant from a clinical perspective. In the revision, we include collagen measurements by ELISA that show increased levels in transgenic macrophages, suggesting that the changes in RNA translate into changes in protein (Fig. 6F). However, these data need to be taken with caution. The increase in protein levels, although significant, is moderate, and it is possible that the increase in collagen expression primarily reflects the induction of an alternatively activated macrophage phenotype that is less inflammatory and exhibits reparative features. Given that LDAH also inhibits MMPs, the induction of fibrosis in lesions might involve a combination of collagenogenic and anti-collagenolytic effects. This has been discussed.

Reviewer #3 (Remarks to the Author):

The interesting manuscript by Paul and colleagues demonstrates that LDAH modulates macrophage activation and protects against atherosclerosis through lipolytic mobilization of endogenous LXR sterol ligands. The authors used LDAH transgenic and knockout mouse models to investigate the role of LDAH in foam cell formation and atherogenesis and identified sterol esters as potential substrates of LDAH using lipidomic analyses of primary macrophages and atherosclerotic lesions. In contrast to the most potent synthetic LXR agonists, the induction of LXR target genes by LDAH was not paralleled by induction of lipogenic genes, which previously argued against the clinical use of LXR agonists. Most of the results support the conclusions and claims, only the quantification of macrophage and collagen content in the plaques needs further clarity.

Major concerns:

Figure 2A, B: It is unclear how exactly the distinct plaque components (e.g. macrophages, necrotic core, and collagen) were quantified as their relative quantification of the respective areas (macrophage + necrotic core + collagen) sum to > 100%. Mac3 staining, especially in Fig. 2B, appears to be unspecific as no cells (nuclei) are visible. Please provide a negative staining control. The same is true for Figure 3 F,G, where macrophage plus collagen areas sum to > 100%. In the section from Tg mice (Fig. 3G), the yellow line that was supposed to delineate collagen and serve for its quantification also includes macrophages and necrotic core. Were these parts also included to determine collagen area? These issues need to be addressed and clarified.

We apologize for the confusion. All lesional parameters were measured specifically in lesion areas, i.e. the media layer was not included in the measurements. The yellow line in collagen images is meant to differentiate collagen within the lesion that was specifically measured from arterial wall media collagen that was not measured, and this has been stressed in the legends. All parts of the lesion areas were included for the collagen measurements.

To address the reviewer's comments, we have:

- 1) Replaced the Mac3 images in Fig. 2B with other images with better nuclei staining.
- 2) Re-quantified all parameters in this figure. The quantification is very similar to the original. In our opinion the >100% of lesion area is due to co-staining in same regions and is a reflect of the pro-fibrotic effects of LDAH.

- 3) Given that the figures are already very complex, we have added consecutive sections with negative staining controls to Fig. S5 (also see below). The immunostaining was performed in

parallel and exactly as we did with the positive staining, except for the inclusion of the primary antibody. All other steps, including blocking and washes, second AB, incubation with ABC reagent, NovaRed peroxidase substrate, and counterstaining with hematoxylin, were identical.

Figure 2: The title states less necrotic core, but this was not determined in males. However, it should be included because it was also indicated in the graphical summary.

A panel with measurements of necrotic cores in lesions of males has been included in the revised Figure 2 (also see below). As usual in ApoE-KO mice, lesions of males are less advanced than in females, and males' necrotic cores were still very scarce in both genotypes and there were no differences between genotypes. These results have been discussed in the 'Results' section'

Figure 1B-E and throughout the manuscript when Apo^{-/-} are compared to Ldah-tg/ApoE^{-/-} mice: The authors are urged to change the misleading labeling WT and Tg in all figures. In addition, the figure legend to Fig. 1 needs to indicate that Apo^{-/-} and Ldah-tg/ApoE^{-/-} mice are being compared.

Thank you. We have replaced the labeling in the figures and improved the figure legends to more clearly indicate that comparisons were performed between *Ldah^{0/0}ApoE^{-/-}* (WT) and *Ldah^{Tg/0}ApoE^{-/-}* (Tg) littermates, or between *Ldah^{+/+}ApoE^{-/-}* (WT) vs. *Ldah^{-/-}ApoE^{-/-}* (KO) littermates. Examples of the new labeling can be seen in the two previous figures.

Minor concerns:

Introduction: Ref 2 might be deleted because the definition of foam cells is not the main message of this publication and since foam cells in general do not require citation.

We have removed this reference.

Usually, *** indicates $p < 0.001$. Why did the authors choose *** $p < 0.005$?

It was arbitrarily chosen. But this is a good point, and we have replaced the numbers of asterisks by the exact format returned by the GraphPad Prism software through the manuscript.

In the revised manuscript, for all figures:

* $p < 0.05$

** $P < 0.01$

*** $p < 0.001$.

**** $p < 0.0001$

Please indicate in the figure legends in which subfigures the t-test or Mann-Whitney U test was applied, respectively.

We have tried to incorporate as much statistical information as reasonably possible in the figure legends. However, some of the figures summarize very large datasets, for example the lipidomics data in Figures 4 and 7. In these figures the statistical information is more generic. However, given that Nature Communications requires a 'Data Source' file, in addition to the information in the legends, for each parameter in the data source file we have added detailed statistic information, including Shapiro-Wilk tests to test normality, F tests to test variances, and the pertinent statistical test (t-tests, Welch's t-tests or Mann U tests). All those were two-tailed.

Figure 3F,G: Figure legends are swapped.

Thank you!... Yes, they were swapped. We have corrected this pitfall.

Figure S2: Why did the authors perform a two-tailed t-test? Welch correction seems more appropriate because the SDs are different.

Variances in Figure S2 were not different, and therefore the data were analyzed by two-tailed t-test. However, as with most of the data in this manuscript, t-test and Welch's t-test would have delivered a very similar message. For example, the p-values of 'collagen area' in Figure 2S are 0.0032 and 0.0050 by two-tailed t-test and Welch's t-test, respectively. For the panel of 'intensity/area' the p values are 0.0043 (t-test) and 0.0059 (Welch's).

REVIEWER COMMENTS

Reviewer #1 (Remarks to the Author):

The authors have adequately addressed my concerns.

Reviewer #2 (Remarks to the Author):

I sincerely thank the authors for their efforts in addressing my various observations. The issues regarding the analysis of single-cell RNA-seq data, collagen production by macrophages, and the quantification of desmosterol have been addressed satisfactorily.

Concerning the major point about LXR regulation, which underpins the mechanism proposed by the authors, I find the responses provided less convincing. There is indeed a noticeable discordance between the RNA-seq data and the QPCR results, and I appreciate the authors for acknowledging this discrepancy. Furthermore, the analysis of existing transcriptomic data does not seem fully convincing. Utilizing data published in two studies (PMID: 30602495; 22292898), the authors demonstrate that 98 genes modulated in LDAH-TG are LXR targets. However, among these 98 genes, approximately half are induced and the other half repressed, with many major LXR targets absent from this list. It would be essential to clarify the criteria used to define the lists of genes regulated by LXR in these studies, including how many genes in total and the enrichment rate compared to the genes regulated in LDHAH-TG, to determine if this enrichment is indeed significant.

If it appears that LDAH regulates ABCA1 and ABCG1 to some extent and represses pathways involved in lipogenesis, potentially via the inhibition of SREBP pathways, I would be cautious about asserting that global activation of LXR pathways is the primary mechanism involved in the atheroprotective effects of LDHAH. I suggest downplaying this point in the manuscript.

Reviewer #3 (Remarks to the Author):

The authors have addressed all the reviewers' concerns in detail.

We thank the reviewers again for taking the time to review this manuscript. We are glad that most of the previous criticisms have been addressed satisfactorily and in detail. However, some concerns remain, particularly the need to downplay the assertion of LXR pathways as the primary mechanism of LDAH atheroprotection. The following is a point-by-point response to the reviewers' comments:

REVIEWER COMMENTS

Reviewer #1 (Remarks to the Author):

The authors have adequately addressed my concerns.

Thank you very much for taking the time to review the manuscript and for the insightful critique.

Reviewer #2 (Remarks to the Author):

I sincerely thank the authors for their efforts in addressing my various observations. The issues regarding the analysis of single-cell RNA-seq data, collagen production by macrophages, and the quantification of desmosterol have been addressed satisfactorily.

We also would like to thank the reviewer for raising these meaningful comments, which significantly improved the manuscript and better aligned this research with the recent literature in the field.

Concerning the major point about LXR regulation, which underpins the mechanism proposed by the authors, I find the responses provided less convincing. There is indeed a noticeable discordance between the RNA-seq data and the QPCR results, and I appreciate the authors for acknowledging this discrepancy. Furthermore, the analysis of existing transcriptomic data does not seem fully convincing. Utilizing data published in two studies (PMID: 30602495; 22292898), the authors demonstrate that 98 genes modulated in LDAH-TG are LXR targets. However, among these 98 genes, approximately half are induced and the other half repressed, with many major LXR targets absent from this list. It would be essential to clarify the criteria used to define the lists of genes regulated by LXR in these studies, including how many genes in total and the enrichment rate compared to the genes regulated in LDHAHTG, to determine if this enrichment is indeed significant.

If it appears that LDAH regulates ABCA1 and ABCG1 to some extent and represses pathways involved in lipogenesis, potentially via the inhibition of SREBP pathways, I would be cautious about asserting that global activation of LXR pathways is the primary mechanism involved in the atheroprotective effects of LDAH. I suggest downplaying this point in the manuscript.

In the methods section we provide additional information on the criteria used to define the lists of genes regulated by LXR based on published papers that performed genome-wide ChIP-seq and transcript profiling in macrophages, resulting in identification of 2018 LXR target genes in total that were activated or repressed by synthetic LXR agonists. We compared these with 872 genes modulated in LDAH-Tg and identified 98 overlapping genes. Hypergeometric probability testing based on a total of 24214 protein encoding genes (per JAX/MGI websites https://www.informatics.jax.org/mgihome/homepages/stats/all_stats.shtml), show significant enrichment of the 98 overlapping genes ($p < 0.001$, enrichment 1.3). The notion that some of LDAH's effects are LXR-dependent is also supported by the LXR loss-of-function and rescue experiments in Figure 6. However, there is no question that the reviewer has raised a very valid point, and LDAH may also protect through LXR-independent mechanisms. Thus, we have edited the text to underscore that foam cells accumulate many lipid substrates, that regulatory sterols display other biological functions and, in general, to downplay the role of global LXR pathway activation as the central mechanism of atheroprotection by LDAH.

Some specific examples:

- The last paragraph of the discussion (lines ~453-457 in the track changes file) reads as follows:

'... foam cells accumulate hundreds of lipid metabolites, and it is possible that some of LDAH's mechanisms of atheroprotection are related to the metabolism of other substrates that we have not been able to identify. Alternatively, in addition to LXR activation, regulatory sterols display other biological functions that could also affect atherosclerosis development...'

- Lines 404-406 in the discussion have been edited to downplay LXR sterol ligands be more inclusive about lipids in general:

Before: *'... like certain sterol LXR ligands, could counter other pro-inflammatory signals of lipoproteins. Given that in foam cells and atheroma these lipids are also abundantly stored as esters, mobilization from LD compartments could significantly enhance their protective actions...'*

After: *'...Given that many of the lipids that accumulate in foam cells can be stored as esters in cytoplasmic LDs, the trafficking in and out of LD compartments could significantly modulate the effects of bioactive species...'*

- Lines ~294-295, in results, after LXR loss of function experiments, a new sentence also mentions that there might be other mechanisms: '*...these studies do not exclude other potential protective mechanisms...*'.
- Other changes have been incorporated through the manuscript, for example, by changing '*natural LXR sterol ligands*' to '*bioactive sterols*', and similar.

Reviewer #3 (Remarks to the Author):

The authors have addressed all the reviewers' concerns in detail.

Thank you veery much for taking the time to review the manuscript and for the insightful critique.

REVIEWERS' COMMENTS

Reviewer #2 (Remarks to the Author):

I have no further comments.
The authors adressed all my concerns